# Meteorological drivers of resource adequacy failures in current and high renewable Western U.S. power systems

Srihari Sundar [1] ✉, Michael T. Craig [2,3] ✉, Ashley E. Payne[4], David J. Brayshaw[5] & Flavio Lehner [6,7,8]

Power system resource adequacy (RA), or its ability to continually balance energy supply and demand, underpins human and economic health. How meteorology affects RA and RA failures, particularly with increasing penetrations of renewables, is poorly understood. We characterize large-scale circulation patterns that drive RA failures in the Western U.S. at increasing wind and solar penetrations by integrating power system and synoptic meteorology methods. At up to 60% renewable penetration and across analyzed weather years, three high pressure patterns drive nearly all RA failures. The highest pressure anomaly is the dominant driver, accounting for 20-100% of risk hours and 43-100% of cumulative risk at 60% renewable penetration. The three high pressure patterns exhibit positive surface temperature anomalies, mixed surface solar radiation anomalies, and negative wind speed anomalies across our region, which collectively increase demand and decrease supply. Our characterized meteorological drivers align with meteorology during the California 2020 rolling blackouts, indicating continued vulnerability of power systems to these impactful weather patterns as renewables grow.

Access to reliable, or uninterrupted, and low-cost electricity underpins human health, and well-being[1]. Designing a reliable system while minimizing costs is the central objective of power system planning[2]. Reliability partly depends on maintaining resource adequacy (RA), which is the system's ability to continually balance electricity supply (or generation) and demand despite the occurrence of unexpected events[3]. RA failures, i.e., times where demand exceeds supply operationally at bulk power systems (BPS) level, are often responsible for large-scale rolling outages, e.g., in California in 2020[4] and Texas[5] in 2021. These two events were caused by a combination of higher than anticipated demand, due to a heatwave (in CA) and a cold snap (in TX), and generator outages driven by extreme weather. This necessitated

intervention, like rolling outages, from the system operator to prevent catastrophic consequences to the system.

Meteorology affects RA through effects on electricity supply and demand. In BPS dominated by thermal electricity generators, surface air temperature is the main meteorological driver of supply and demand. Low and high surface air temperatures affect demand through increased use of building heating, ventilation, and air conditioning (HVAC) for heating and cooling, respectively[6,7]. Surface air temperature also affects supply. Specifically, extreme heat increases deratings of thermal power plants[8,9] and solar photovoltaics, while extreme cold and heat increases forced outage rates of thermal and hydroelectric power plants[10].

[1]Department of Aerospace Engineering, University of Michigan, Ann Arbor, MI, USA. [2]School for Environment and Sustainability, University of Michigan, Ann Arbor, MI, USA. [3]Department of Industrial and Operations Engineering, University of Michigan, Ann Arbor, MI, USA. [4]Tomorrow.io, Boston, MA, USA. [5]Department of Meteorology, University of Reading, Reading, UK. [6]Department of Earth and Atmospheric Sciences, Cornell University, Ithaca, NY, USA. [7]Climate and Global Dynamics Laboratory, National Center for Atmospheric Research, Boulder, CO, USA. [8]Polar Bears International, Bozeman, MT, USA. ✉e-mail: sriharis@umich.edu; mtcraig@umich.edu

Two trends complicate the link between meteorology and RA: (1) increasing penetrations of wind and solar power, and (2) non-stationary meteorology driven by natural variability and anthropogenic climate change. Since wind and solar power are a function of wind speeds and solar irradiance, increasing wind and solar power penetrations will increasingly link electricity supply to these meteorological variables. Wind speeds and solar irradiance exhibit significant spatio-temporal variability[11,12] and forecast and projection uncertainty[13,14], complicating RA assessment. Non-stationary meteorology driven by intensifying climate change further complicates RA assessment. As historical meteorology becomes increasingly non-representative of future meteorology, RA assessment of future system fleets will need to increasingly rely on projected future meteorological timeseries to account for the transient nature of the current climate state. However, generating high-quality meteorological projections that account for climate change remains an active area of research limited by methodological uncertainties, and computational power[15]. Generating high-quality future meteorological timeseries is especially challenging at the high spatio-temporal resolution (e.g., hourly) typically required for RA analyses[16].

In response to these challenges, this paper aims to better understand the meteorological drivers of RA, focusing specifically on RA failures, and how increasing renewable generation affects those drivers. Better understanding these relationships is crucial for several reasons. First, the meteorology that drives (and co-occurs with) RA failures will determine human health impacts, which can be highly heterogeneous across space and socioeconomic groups[17]. Better understanding the link between decarbonization and drivers of RA failures can shed light on investment needs in BPS and communities to mitigate possible health impacts and achieve more equitable outcomes. Second, characterization of historic meteorological drivers can guide in evaluating, selecting, and downscaling general circulation models, which is essential for making informed adaptation investments in the power sector[18,19]. Third, once meteorological drivers of RA failures are characterized, long-range probabilistic forecasting at the subseasonal to seasonal scale can act as a more informed early warning system for system operators and emergency preparedness organizations[20].

We characterize meteorological drivers of RA failures using weather regimes. Weather regimes represent atmospheric circulation as belonging to a finite number of states or patterns[21,22]. These states are constructed by applying clustering techniques to variables representing large-scale atmospheric flows, e.g., geopotential height. The resulting large-scale patterns have strong associations with surface-level meteorological variables that directly affect the power system, including extreme surface air temperatures[23–25]. These patterns indicate several processes like temperature advection and subsidence which can, under certain conditions, drive extreme events in the power system. The patterns persist over large spatial and temporal scales, and unlike the high-frequency variations exhibited by surface meteorology, the patterns' spatio-temporal variations are better captured by general circulation models (GCMs). Previous research has sought to link the changes in frequency and return periods of these large-scale patterns with the occurrence of extreme events under a changing climate using data from GCMs[26–29]. The spatial coverage of these large-scale atmospheric circulation patterns makes them valuable analogs for surface meteorology over large geographic regions. Using these synoptic drivers in planning and operations can benefit system operators when thinking about RA due to current and future systems' increasing dependence on generation over larger areas and interconnected balancing authorities.

Our research contributes to two literatures. The first literature analyzes meteorological drivers in the power system, but does not consider RA, a gap that we fill. Within this set, a few studies examine meteorological drivers of periods of low renewable generation or high net demand (demand minus renewable generation)[30–33]. Meteorological drivers in these papers include surface meteorology and atmospheric circulation during these periods. Further, other studies describe weather regimes as drivers of renewable generation, variability, and net demand in the European power system[34–36]. The second literature analyzes RA, but does not consider meteorological drivers, a gap that we also fill. In this broad RA umbrella, studies quantify the effect of using different RA metrics on reserve procurement decisions[37] and capacity values[38]. Other studies quantify the contribution of generators[39,40] and transmission[41] to RA. A final group of studies quantify system RA under changing generator and/or weather. For instance, Turner et al.[42] quantify RA changes (in probability and magnitude) driven by decarbonization decisions and climate change impacts on electricity demand and hydropower generation in the Pacific Northwest.

To address these gaps, we answer the following research questions: What large-scale circulation patterns drive risk of regional resource adequacy failures? And how do these drivers change with increasing wind and solar penetrations? We define resource adequacy (RA) as the ability of a power system to continually balance electricity supply and demand[3], and quantify RA on a probabilistic, hour-to-hour operational basis. We conduct our study for the U.S. Western Electricity Coordinating Council (WECC) footprint given its rapid growth in wind and solar penetrations, aggressive wind and solar targets, and recent resource adequacy failure[43]. Using a one-way impact analysis that decides fleet investment to meet the standard resource adequacy target (1 day in 10 years), identifies resource adequacy failures, and finds meteorological drivers of these failures for increasing renewables penetrations, our research is the first to link weather patterns and power systems operations in the United States, and the first to characterize weather regimes driving RA failures.

Our analytical pipeline uses methods from power system and synoptic meteorology domains (Fig. 1). We first construct fleets that generate increasing levels of wind and solar electricity (hereafter renewable electricity or RE) using a capacity expansion model (CEM) (see Methods "Capacity expansion"). The CEM is a deterministic linear program that minimizes total system cost, which is the sum of the cost of new capacity investments and the cost of electricity generation of existing and new units. The cost of electricity generation is the sum of fixed operations and maintenance (O&M) costs and variable electricity generation costs, which include fuel costs and variable O&M costs. The CEM specifically optimizes new investments in wind, solar, 4-h electricity storage facilities, inter-regional transmission capacities, and operations of existing and new units, and inter-regional electricity flows. The CEM does not optimize investment in new thermal facilities given its coupling with our RAM, which adds or removes thermal facilities to reach a given reliability target. Investment and operational decisions are subject to numerous generator- and system-level constraints, including hourly balance of supply and demand and electricity flows, limited inter-regional electricity flows, hourly site-specific wind and solar resource availability, engineering and economic-based unit operations, and limited technology-specific investments. To capture co-variability and extremes in electricity demand and wind and solar generation, we use observed hourly electricity demand for WECC[44] and coincident spatially-differentiated RE capacity factors (see Methods "Data description"). In our models we divide WECC into five constituent sub-regions, as used by WECC in its Western Assessment of Resource Adequacy report (ref. SI Fig. A.3)[45]. Between each pair of sub-regions, we model transmission flows using the transport method, which caps hourly inter-regional electricity flows between sub-regions to a fixed transmission capacity. Investment decisions in storage, occur at the five-region level; in transmission, between each pair of regions; and in wind and solar, at spatially-differentiated resource locations on a roughly 30 by 30 km grid. RE penetration levels are enforced at the WECC scale.

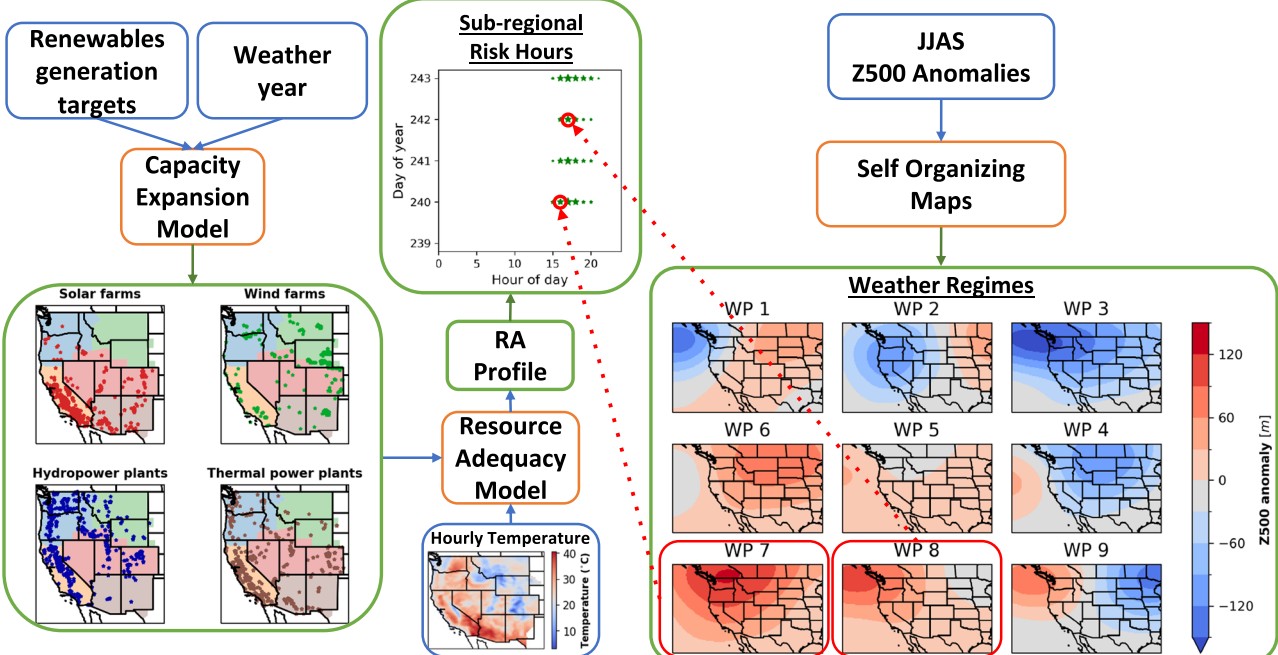

**Fig. 1 | Analytical pipeline.** We use a capacity expansion model (CEM) to construct generator fleets with increasing renewable penetrations and different weather years. Maps show the sizes and locations of facilities for 60% renewables penetration and 2019 weather. These fleets are input into a resource adequacy model (RAM) to quantify hourly loss of load profiles (LOLPs), yielding a resource adequacy (RA) profile (in this figure we only represent the RA risk hours). We then map the risk hours in the RA profile to weather regimes, which we identify with self-organizing maps (SOMs) applied to 500 hPa geopotential height (Z500) anomalies. Depicted weather regimes are the SOM outputs for extended summer months, with positive anomalies (high pressure systems) in the bottom left and negative anomalies (low pressure systems) in the top right. By varying renewable penetrations and weather years, we characterize meteorological drivers of risk hours. Red arrows depicting attribution of risk hours to weather regimes is for illustrative purposes only.

We then quantify a RA profile for each fleet and each sub-region from the CEM using a resource adequacy model (RAM), which simulates stochastic forced outages of generators using a non-sequential Monte Carlo sampling procedure and finds hours where there is a non-zero probability of demand exceeding total available generation (see Methods "Resource adequacy model"). We use empirically-derived temperature-dependent forced outage rates for NGCC and hydropower facilities, constant outage rates for other generators, and do not account for outages in storage units[10,46]. Storage assets are dispatched on a chronological hourly basis within the RA model within each Monte Carlo iteration after dispatching all the other generators using a greedy dispatch policy[39,47]. From the RAM, we obtain a timeseries of loss of load probabilities (LOLPs) by hour of the year, which we refer to as the RA profile. This RA profile is a function of short-term operations from the RAM. Hours with LOLPs greater than zero indicate a risk of an RA failure; we refer to these hours as RA risk hours or risk hours.

Finally, to characterize the meteorological drivers of RA failure, we map the 500 hPa geopotential height (Z500) anomalies in these risk hours to the western US summer weather regimes. These regimes are constructed based on June–September daily Z500 anomalies from a 40 year period using self-organizing maps (SOM), and each regime is represented by a characteristic weather pattern (WP) (see Methods "Meteorological analysis"). The characteristic WPs show regimes with varying Z500 anomalies over the region, ranging from positive anomalies (high pressure systems, WP7) to negative anomalies (low pressure systems, WP3) (Fig. 1 weather regimes panel). Each weather regime produces different surface weather patterns, e.g., high pressure anomalies in WPs 7 and 8 drive extreme heat events across the Western US, as later illustrated in our results. The WPs corresponding to regimes identified based on the risk hours characterize the large-scale patterns contributing to RA failures. By running this integrated modeling framework for four weather years (2016 through 2019) and

RE penetrations (Current, 30%, 45%, and 60%, see section "Scenarios" for definition of RE penetration), we quantify the effect of increasing renewables on meteorological drivers of RA and the robustness of this effect across independent weather years. While using four weather years does not sample the full distribution of possible weather events and associated impacts on RA and RA failures, it does cover over 35,000 hours and permits us to use observed hourly electricity demand with coincidental wind and solar generation.

Using this analytical pipeline, in this work, we show that RA failures in WECC are driven by WPs corresponding to high pressure anomalies (WPs 6, 7, and 8 in Fig. 1) over the region. These WPs correspond to high surface air temperatures and low wind speeds across WECC and with low solar irradiance in large areas with solar PV facilities. These meteorological conditions cause compounding impacts on electricity supply and demand, ultimately resulting in risk of resource inadequacy (i.e., RA failures). As renewable penetrations increase, the risk of RA failures increasingly concentrates within the WP with the highest pressure anomaly (WP 7).

## Results

We divide our results into two sections. First, we quantify the effect of increasing renewable penetrations on meteorological drivers of risk hours for a single weather year (2019). Second, we repeat this analysis to characterize meteorological drivers of risk hours across multiple weather years at increasing renewable penetrations. We restrict our analysis to the CAMX region for two reasons. First, NERC's Long-Term Reliability Assessment (LTRA) indicates CAMX is the most vulnerable WECC region to resource adequacy failures in the near term, with LOLH of 0.72 and 9.79 in 2024 and 2026 respectively in the 2022 assessment. By comparison, other regions in WECC have LOLH of up to 0.03 (2024) and 0.37 (2026), an order of magnitude less than CAMX. Thus, understanding meteorological drivers of RA failures in CAMX

can provide significant near-term value to decision makers and serve as a model for analyses in future regions. Our resource adequacy results agree with the LTRA, as we find CAMX has at least 4x and 27x more probability of resource adequacy failure than any other WECC region in the current and RE penetration greater than 30% fleets respectively across the years. Second, we find that in all but one scenario we analyze, and in all RE penetration greater than or equal to 30%, the CAMX risk hours coincide with risk hours in other regions if failures occur in other regions. Across the weather years, the current fleets correspond to a RE penetration ranging from 9–9.4%, so we denote these fleets as 9% RE penetration in our results.

## Meteorological drivers under increasing renewable penetrations for the 2019 weather year

Using our CEM, we construct generator fleets in which RE generation accounts for increasing percentages of annual demand. As renewable penetrations increase from 9% (or current levels) to 60% of annual demand, wind, solar, and storage capacities (at the interconnection level) increase from 20 GW, 16 GW, 5 GW to 103 GW, 70 GW, and 7 GW respectively, while NGCC capacities decrease from 49 GW to 35 GW (Fig. 2, see SI Fig. A.8 for sub-regional regional capacities). Figure 3 depicts each system's RA profile by showing the magnitude of hourly LOLP and timing of risk hours. Across renewable penetrations, all risk hours occur in the extended summer months (i.e., June through September or JJAS). Most risk hours occur between 4 and 8 p.m. Pacific Standard Time (PST). As renewable penetrations increase from 9% to 60%, the number of risk hours decrease from 68 to 10 and increasingly concentrate into the period between 6 and 8 p.m. PST. The decrease in risk hours is driven by increasing available generation in many hours of the year, including in hours that previously had low LOLPs. In these hours, increasing available generation results from wind and solar capacity increases exceeding NGCC capacity decreases. Particularly, the increasing storage capacity reduces risk in the early evenings. As risk hours decrease, hourly LOLPs increase. For instance, as renewable penetrations increase from 9% to 60%, maximum LOLPs increase from 0.27 to 0.63 (SI Fig. A.9b).

To attribute RA failures to WPs, we map each risk hour to the prevailing weather regime, then quantify the number of risk hours and cumulative LOLP in each regime (Fig. 4). The cumulative LOLP equals

the sum of LOLPs across hours mapped to a given weather regime, so is a function of the number of risk hours in a given weather regime and the LOLP in each of those hours. The cumulative LOLP also equals the expected loss of load hours (LOLH) attributed to each regime. Using either number of risk hours or cumulative LOLP metrics, WPs 6, 7, and 8 predominantly drive RA failures across renewable penetrations (Fig. 4). These WPs correspond to high pressure anomalies that cover the entire Western US (as shown in Fig. 1). Of those WPs, WP 8 accounts for most RA failures, e.g., 39–50% of risk hours and 54–82% of cumulative LOLP across renewable penetrations.

The relative importance of WPs in driving RA failures is robust across increasing renewable penetrations for the 2019 weather year. As renewable penetrations increase from 9% to 60%, the number of risk hours driven by WP 8 decrease from 27 to 5, respectively, while the numbers of risk hours driven by WPs 6 and 7 exhibit an overall decrease, from 21 to 3 and from 19 to 2, respectively. Increasing renewable penetration has the opposite effect on cumulative LOLP driven by WPs 7 and 8. As renewable penetrations increase from 9% to 60%, the cumulative LOLP driven by WP 7 increases from 0.3 to 1.1, whereas cumulative LOLP driven by WP 8 decreases from 1.2 to 0.9 (Fig. 4). Cumulative LOLP driven by WP 6 shows an overall decrease from 0.7 to 0.5 comparing 9 and 60% renewable penetrations.

Mechanistically, surface meteorology, not high-pressure anomalies in the middle atmosphere, impact power system RA. To understand how the high pressure anomalies in WPs 6, 7, and 8 drive RA failures, we analyze surface meteorology corresponding to each weather regime (ref. Methods "Meteorological analysis"). We find that these WPs correspond to positive surface temperature anomalies, and mixed surface solar radiation and wind speed anomalies across large regions of WECC (Fig. 5). Positive temperature anomalies lead to higher than average generator forced outages and demand. Concurrently, negative and low positive solar radiation anomalies lead to lower than average solar generation. While surface solar radiation anomalies are not negative across WECC in the 3 impactful weather patterns, in WP 7, these anomalies are negative in the CAMX region where a large fraction of solar capacity is installed (Fig. 1). WPs 6, 7, and 8 also exhibit negative wind speed anomalies in large portions of the western US, and more notably so in WP 7. Each of these WPs include surface meteorology anomalies that reduce RA at low and high renewable penetrations, explaining the robustness of these three WPs in driving most RA failures at renewable penetrations ranging from 9 to 60%. Of these three WPs, WP 7 increasingly drives total risk with increasing RE penetrations as it has the large positive temperature anomalies, largest negative solar anomaly over the Southwest, and largest negative wind speed anomaly over the entire region. Other WPs do not exhibit the same combination of surface temperature, wind speed, and solar radiation anomalies that WPs 6, 7, and 8 do, explaining their relative unimportance in driving RA failures.

## Meteorological drivers across different weather years

The above discussion examines drivers of RA failures across renewable penetrations for a single weather year, 2019. Given significant inter-annual variability in meteorology and climate, we repeat our above analysis across four weather years (2016 through 2019) or the duration of our combined data timeseries. This approach treats each meteorological year as an independent observation, allowing us to quantify the robustness of our results to different weather years.

Across weather years and RE penetrations, NGCC and wind capacities output by the CEM do not significantly differ across years. For instance, at 60% renewable penetration, NGCC capacities range from 45 to 35 GW, and wind capacities range from 95 to 116 GW across weather years (SI Fig. A.9a). Solar capacities exhibit a larger range across weather years, e.g., ranging from 27 GW in 2017 to 70 GW in 2019 at 60% RE penetration, with low solar capacity coinciding with high NGCC capacity (Fig. SI. 6a). Storage capacity also exhibits a larger

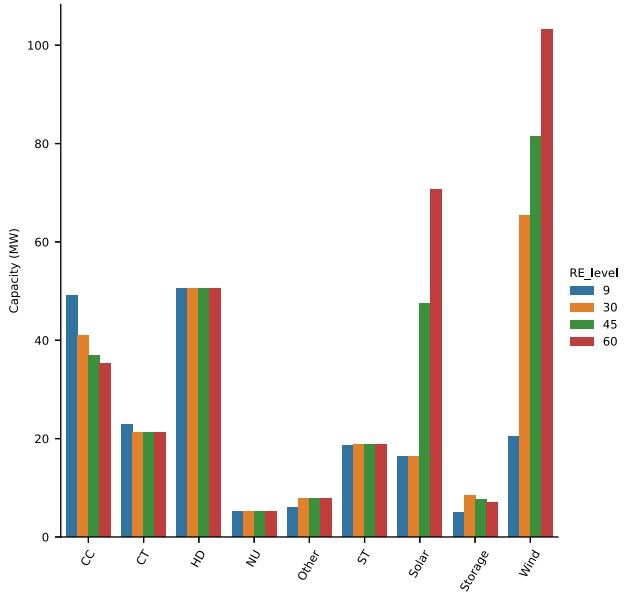

**Fig. 2 | Installed capacities of different generation sources with increasing renewable penetrations for the 2019 weather year.** This figure shows WECC wide total capacities with color bars representing different RE penetrations.

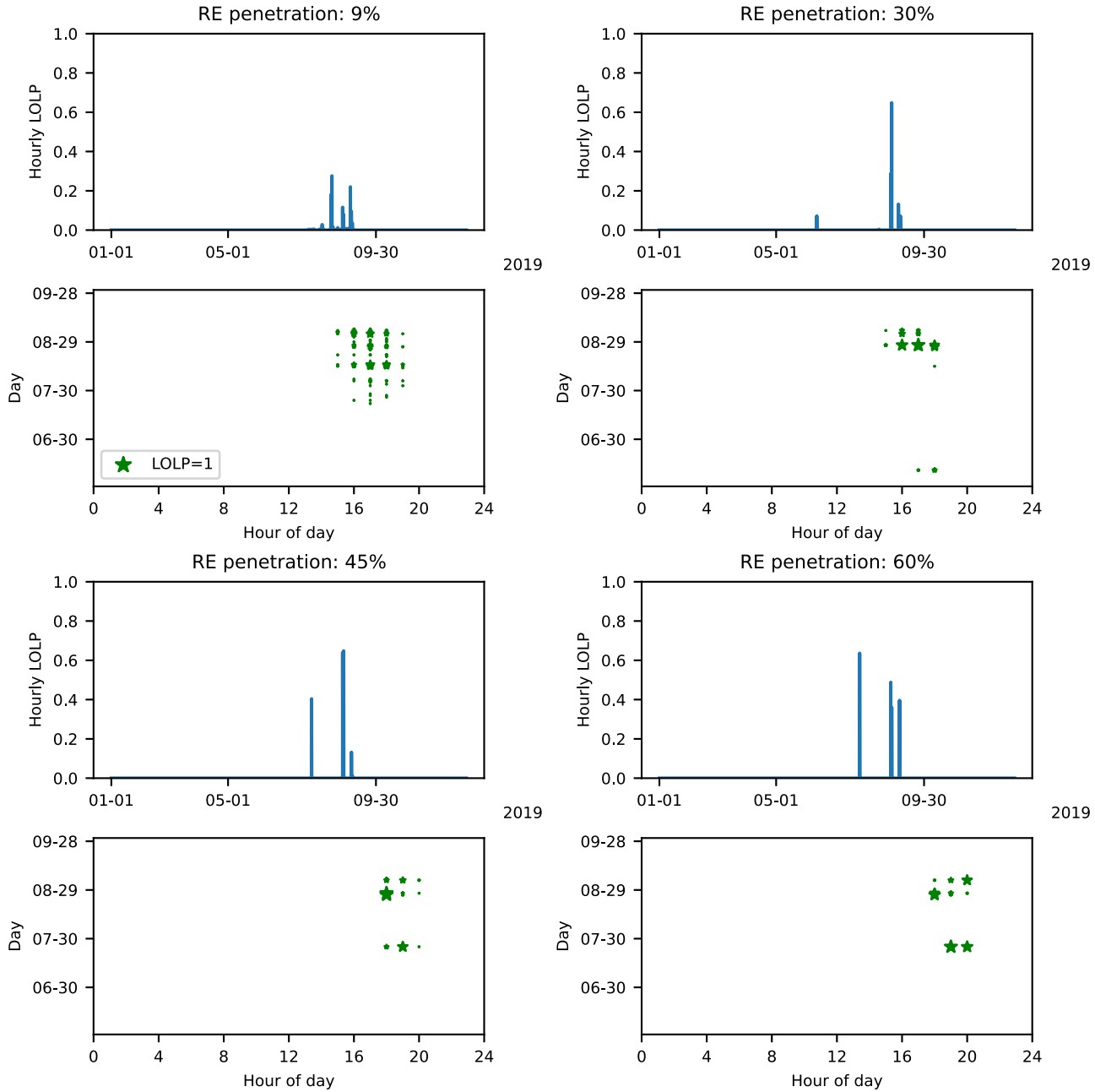

**Fig. 3 | RA profiles and timing of RA failures.** For the 2019 weather year and for each renewable penetration, this figure shows, (i) hourly LOLPs across the entire year (i.e., the RA profile) and (ii) the date and hour of day (in PST) when RA failures occur, where the size of star is proportional to the LOLP and the legend shows marker size for LOLP = 1. An LOLP of 0.1 indicates demand exceeds available capacity in 10% of the 250 simulated trials in the RA model.

range, from 7 GW in 2019 to 19 GW in 2018. Our results regarding the number of risk hours and maximum LOLPs are also largely insensitive to different weather years. Specifically, across weather years, risk hours decrease and maximum LOLPs increase between the current fleet and higher RE penetrations (SI Fig. A.9b). For instance, in 2018, risk hours decrease from 53 to 5 and maximum LOLPs increase from 0.3 to 0.96 when renewable penetrations increase from 9 to 60%. For all the weather years and renewable penetrations, we also simultaneously calculate the expected unserved energy (EUE). This is the sum of expected shortfall (in GWh) during each risk hour. SI Fig. A.10 shows the EUE for the different systems with the effective shortfalls ranging from 3.5 GWh to 4.6 GWh and 1.1 GWH to 3 GWh at 9% and 60% RE penetrations respectively.

Meteorological drivers of RA failures are also robust to weather years (Fig. 6). WPs 6, 7, and 8, which are high pressure anomalies,

drive most RA failures across all weather years. Collectively, these WPs drive 87 to 100% of all risk hours and 96 to 100% of cumulative LOLP across weather years. Furthermore, WP 7 emerges as an even more dominant driver of RA failures in 2016 through 2018 than in 2019. In weather years 2016 through 2018, WP 7 accounts for cumulative LOLPs of 84 to 100% of the respective scenario's total risk for renewable penetrations of 9 to 60%, compared to 13 to 43% in 2019 (Fig. 6b). When considering all days in the JJAS months, we find that the number of days attributed to the extreme weather patterns (WP 7 and WP 8, but particularly WP 7) are comparable to the number of days attributed to intermediate weather patterns (such as WPs 4, 5, and 6) (SI Fig. A.6). Moreover, among our study years, 2 years have above trend line occurrences of WPs 7 and 8, and 2 years have below trend line occurrences of WP 7. Despite the total number of days in each WP and variability in occurrence frequency among the years

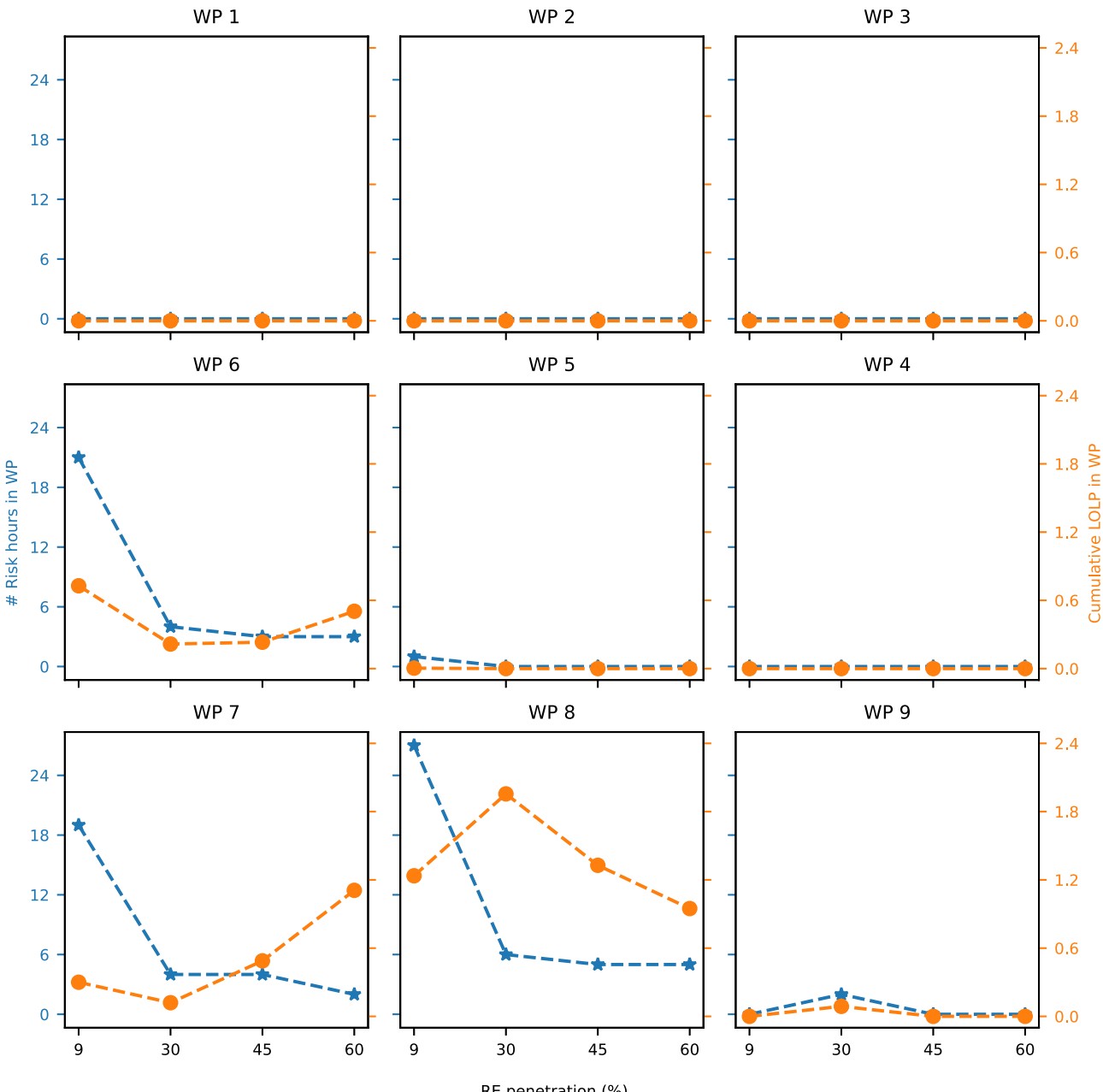

**Fig. 4 | Risk hours and cumulative LOLP attributed to each weather regime in 2019.** For the 2019 weather year, for each renewable penetration this figure shows number of risk hours (blue lines) and cumulative LOLP (orange lines) attributed to each weather regime, where WPs correspond to Fig. 1.

analyzed, WP7 emerges as the more dominant driver at higher RE penetrations across the weather years.

The surface meteorology associated with WPs 6, 7, and 8 in weather years 2016–2018 show similar trends of positive temperature anomalies, negative wind speed anomalies, and mixed solar radiation anomalies in the Southwest as in 2019 (see SI Figs. A11–13). At higher RE penetrations, the risk is attributed to fewer days. So we look at the daily average temperature anomalies for these days (Fig. 7). Though these days are driven by WPs 6, 7, or 8 across the weather years, they represent different distribution of surface meteorological anomalies in the different years. On the RA failure days, the temperature anomalies across these four years show predominantly positive anomalies over large portions of the region, but the magnitude, geographical location and extent of the positive anomalies vary. Some days also exhibit negative anomalies in some regions, but even on these days the anomalies are positive in the California region. SI Figs. A.14 and A.15

show the surface solar radiation and wind speed anomalies for these days.

## Discussion

Maintaining power system RA, and reliability more broadly, faces challenges from evolving supply- and demand-side technologies and non-stationary meteorology. In response to these challenges, this paper characterized meteorological drivers of RA failures by integrating power system and meteorological methods. We found that RA failures in WECC are driven by weather patterns corresponding to high pressure anomalies over the western United States.

The added value that our weather pattern approach gives over just a surface meteorological analysis is that we are able to capture the synoptic scale (1000–2500 km) drivers of the RA failure events. The weather patterns can be used in different ways to incorporate meteorological drivers of the power system in system planning as well

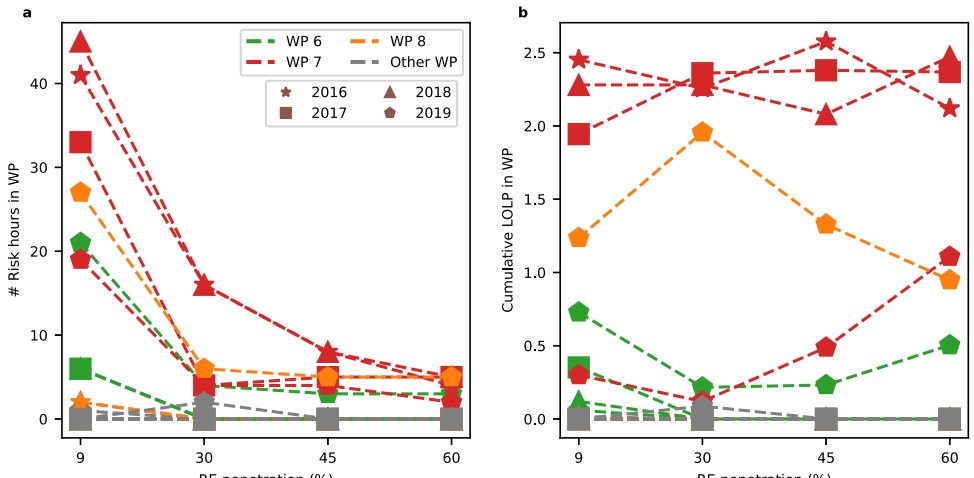

**Fig. 5 | Surface meteorological anomalies corresponding to each weather regime. a** Composites of surface temperature anomalies, **b** surface solar radiation anomalies, and **c** 100 m wind speeds anomalies for the 2019 weather year. The composites are constructed based on the hours from the 2019 extended summer belonging to each weather regime.

**Fig. 6 | Risk hours and cumulative LOLP attributed to each weather regime across all weather years. a** Number of risk hours attributed to each weather regime across the weather years with increasing RE generation levels; **b** Cumulative LOLP attributed to each weather regime across the weather years with increasing RE generation levels.

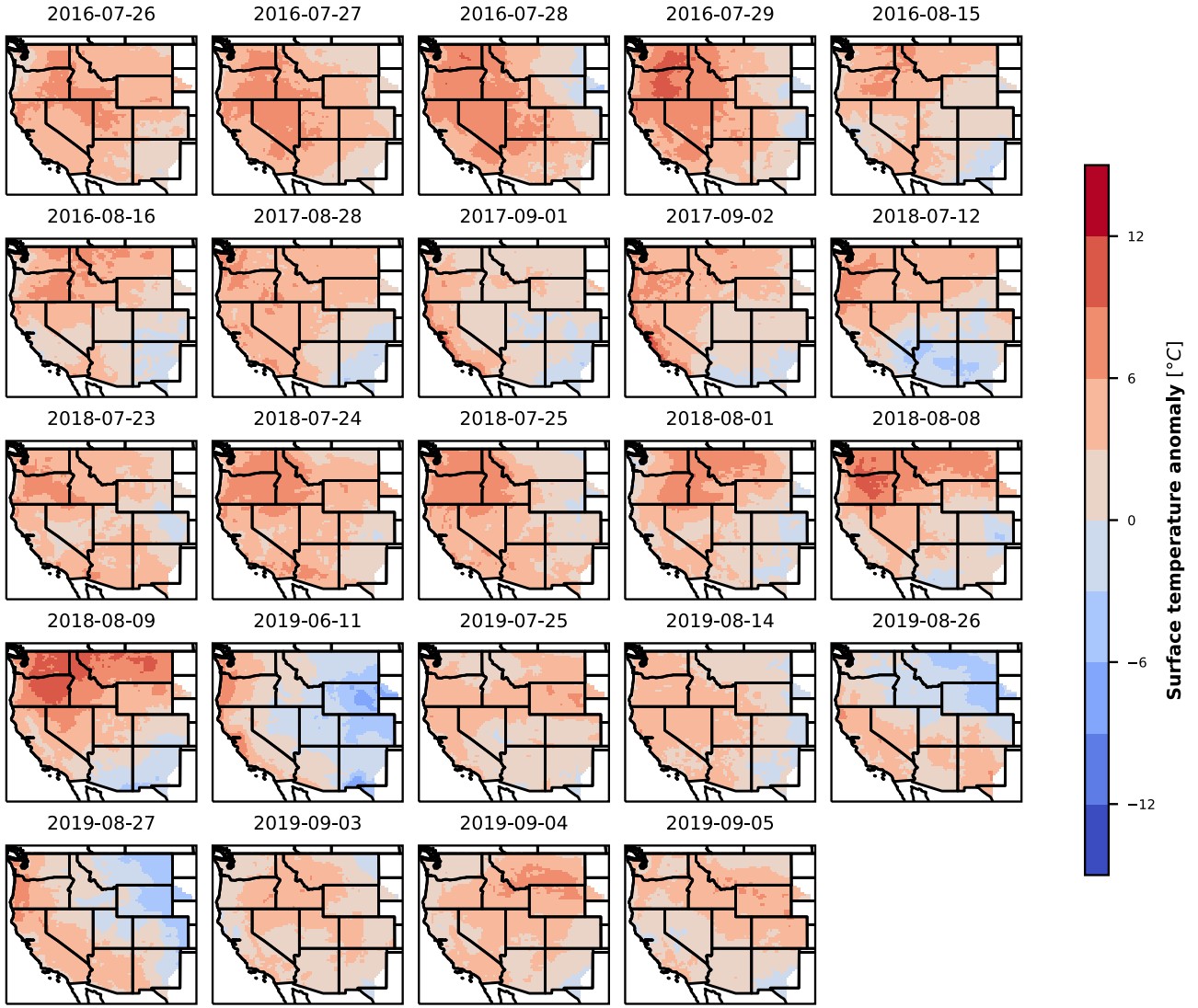

**Fig. 7 | Daily surface temperature anomalies on days with RA failure events for RE penetrations from 30 to 60% across the weather years.** Each panel in this figure shows daily means of surface temperature anomalies on the RA failure days.

as operations, as we move to interconnected continental scale systems. For system planning purposes, current practices mostly involve only using historical meteorological data with techniques like importance subsampling reducing computational costs by providing representative periods to the capacity expansion model[48]. Our findings can improve this subsampling process by providing a physical basis for choosing the representative periods. Further, to make informed investment decisions and maintain system reliability in the future, system planning needs to use future meteorological data from climate projections and the physics based subsampling procedure can help here as well. Future climate projections from global climate models have lower spatial and temporal resolution than required by power system models. Incorporating this future climate data requires computationally costly downscaling[16]. Our methods can reduce downscaling needs and associated costs by guiding selective downscaling of certain time periods of interest, e.g., time periods with high pressure anomalies in the Western US, to drive system planning and operation models. This can help system planners understand further risks, beyond resource adequacy, during these stressful periods. At the operational level, system operators, utilities, power producers, and communities can use the short-term forecasts at the days to weeks timescale and long-range probabilistic forecasting at season-to-season time scale to avoid scheduling maintenance and other related down

times when these patterns are expected to occur. These patterns are characterized by their temporal persistence and ability to represent meteorology at the synoptic scale during the occurrence of extreme events. These characteristics make the WPs more suitable, as an aggregate pointer to capture stressful periods for system operations, than individual surface meteorological variables, which exhibit higher spatio-temporal variations.

Rolling outages in California in the summer of 2020 support our results. On August 14 and 15, the California Independent System Operator (CAISO) instituted rotating electricity outages during an extreme heat storm covering much of the WECC system[4]. These rotating outages were necessitated by higher-than-predicted demand and supply shortages. While we are not able to include 2020 in our analysis due to data limitations, we can analyze atmospheric circulation prevailing during August 14 and 15 using our reanalysis data (Methods "Data description"). We find that the atmospheric circulation on these 2 days exhibits a high pressure anomaly over the Pacific northwest (SI Fig. A.16) and resembles the high pressure WPs in our analysis. Our SOM identifies the circulation pattern on August 14 as belonging to WP 8 and on August 15 as belonging to WP 7. Thus, the CAISO rotating outage event provides real-world evidence for these weather patterns driving RA failures, which we have also identified through our analysis.

While outages threaten human health and well-being regardless of prevailing meteorology, outages during extreme heat can be particularly life threatening[17]. The robustness of high pressure anomalies driving RA failures at renewable penetrations up to 60% suggests that high temperature anomalies will continue to accompany RA failures. Consequences of outages could have disproportionate impacts on vulnerable populations[49], particularly when they align with extreme heat events[50]. Any disparities in outcomes during outages between income groups could widen as upper income individuals increasingly procure distributed energy systems. Our results indicate a long-term need to ensure vulnerable communities have access to potentially lifesaving cooling during outages, e.g., through investing in community hubs at public buildings[51].

Anthropogenic climate change is already affecting weather and climate, including by increasing surface air temperatures across the Western United States[52]. Using the ERA5 reanalysis dataset, we find some evidence for an increase in the frequency of weather regimes with high pressure anomalies from 1981 through 2020 in the extended summer months (SI Fig. A.7). During this period, WPs 7 and 8 (high pressure anomalies over northwest) occur more frequently, while some WPs like 3 and 4 (low pressure anomalies over northwest) occur less frequently. Increasing trend of WP 7 over the last 40 years are statistically significant ($p$ values <0.05) based on a simple linear regression with year as the independent variable and percent of days with the WP as the dependent variable. Specifically, WP7 shows an increase of 0.18 extended summer days per year. Given that we found high pressure anomalies, particularly WP7, drive RA failures, their increasingly frequent occurrence might result in more frequent challenges to maintaining RA. More rigorous analyses are needed to discern and attribute WP trends to aspects of the earth system dynamics, including natural variability versus anthropogenic changes. Emerging research has also found that the change in frequency of certain circulation pattern can compound climate extremes driven by anthropogenic warming[53]. So, better understanding how these impactful WPs will evolve and interact with a changing climate[26] would better inform the risk that climate change poses to RA.

Our research offers several opportunities for extensions. First, to capture co-variability between supply and demand, our analysis is limited to four weather years. To capture long-term climate variability, future research could extend our analysis to multi-decadal timespans using historic data from reanalyses or future data from climate models. Second, future research could also incorporate decarbonization-driven changes on demand including electrification of residential heating and charging of electric vehicles. These extensions face several challenges, though, including estimating electricity demand with bottom-up models and obtaining high spatio-temporal resolution climate model outputs. Third, we do not consider the availability of flexible loads in our models, which can be an avenue for operational adjustments by the system operator to prevent RA failures. Incorporating these demand side changes could reduce the risk in hours with high failure susceptibility. Fourth, in linking specific weather patterns to resource adequacy failures, our research suggests climate downscaling methods designed, trained, and/or validated on these types of weather patterns could be highly valuable in bridging the disconnect between climate and energy system modeling[16]. Additionally, our results suggest RA analyses using future climate data could focus on weather regimes documented here, which could enable a greater computational focus on climate-related uncertainty.

## Methods
### Area of study
Our area of study is the Western Interconnection, which is the region within the continental United States overseen by the Western Electricity Coordinating Council (WECC). We choose the WECC system for its high existing wind and solar installed capacities, its strong wind and solar resources, its large geographic area which makes it susceptible to large-scale meteorology, and its vulnerability to climate change in the near-term. Climate change has already reduced system reliability in WECC, with extreme heat and drought exacerbated by climate change driving outages in California in 2020[4]. We model WECC in terms of its constituent sub-regions in a representation similar to the one WECC uses in its western assessment of resource adequacy report. The five sub-regions are CAMX, Desert Southwest, Northwest Power Pool–Central (NWPP-Central), Northwest Power Pool–Northeast (NWPP-NE), Northwest Power Pool–Northwest (NWPP-NW). Figure A.3 shows the geographic regions which are within the sub-regions[45].

### Capacity expansion
We use a capacity expansion model (CEM) to create future WECC generator fleets that meet increasing renewable generation requirements. We run the CEM for each analyzed weather year, capturing coincident, spatially-resolved meteorology and hydrology for each year. The CEM is a deterministic linear program that minimizes fixed plus variable costs by deciding investment in wind, solar, 4-h utility-scale battery storage, and inter-regional transmission, and operation of existing and new generators, storage, and inter-regional transmission. Wind and solar capacity investment decisions occur at the spatial resolution of our wind and solar resource data, i.e., on a 30 by 30 km grid across WECC, while storage and transmission investments occur at the five-region and inter-regional levels, respectively. Because we couple the CEM with the RAM (described below), which adds or removes thermal generators from each future fleet to meet a given reliability target, we do not add thermal units or retire any existing units in the CEM. Thus, the fleets generated from the CEM form a basis for creating the final fleets used in our analysis. These final fleets are obtained after the RAM adds or removes thermal generators.

The CEM includes numerous system- and generator-level constraints. At the system level, the CEM requires total generation to meet demand in each hour. To approximate system reliability standards, the CEM includes a 13% planning reserve margin, which requires derated capacity to exceed peak demand by at least 13%. Derated capacity accounts for hourly wind and solar generation potential during the peak demand hour, a fixed 5% forced outage rate for wind and solar generators, and for temperature-dependent forced outage rates for all other generator types (see SI Section 3.3 for forced outage rates used)[10]. At the generator level, generation can vary between zero and maximum capacities, following engineering and economic-based unit operations constraints. Wind and solar generation is also limited by hourly, spatially-specific wind and solar capacity factors (see "Data description"). The CEM also decides and constrains hourly charging, discharging, and state of charge of each existing and new storage unit. To examine generator fleets with increasing RE penetrations, the CEM requires total WECC-wide wind plus solar generation to meet a percent of total annual demand (see section "Scenarios" for specification of target levels).

For computational tractability, we run the CEM in hourly intervals for one representative time block per season, with seven sequential days in each time block, and for days with peak annual demand, net demand, and upwards hourly ramp. The representative days capture typical operations and costs, while the peak days capture system capacity and flexibility investment needs. Sampled representative days per season minimize the root mean squared error between sampled and seasonal net demand profiles. Within each time block, the CEM dispatches regional hydropower generation based on historic year-specific generation data.

We formulate the CEM using the General Algebraic Modeling System[54] and solve it using CPLEX[55]. For the full CEM formulation and description, see SI Section 2.

## Resource adequacy model

To quantify resource adequacy on an hourly and annual basis, we combine a Monte Carlo-based non-sequential state sampling procedure with an optimization-based sequential storage dispatch procedure. The state sampling procedure randomly samples forced outages at each generator within every WECC sub-region in each hour of the year 250 times via Monte Carlo simulation (see SI Section 3.2 for justification of sample size). This results in 250 independent capacity curves for the year, each of which are paired with observed hourly demand for the year. Like in the CEM, forced outages are a function of location-specific ambient air temperatures for thermal and hydropower plants[10], are a constant rate of (0.05) for solar and wind plants[46], and are assumed to be zero for storage and transmission (see SI Section 3.3 for forced outage rates used).

Within each sub-region, for each capacity curve after storage dispatch occurs, we identify hours where any sub-region has a loss of load event (where sub-regional demand exceeds available sub-regional generation). For these hours we run a simple network flow optimization problem to determine inter-regional transfers within each Monte Carlo iteration. The optimization objective is to minimize the total cost of energy transfer along the lines and cost of energy not served within the sub-regions, with constraints imposed on line limits and energy available for export from each sub-region (see SI Section 3.1 for transmission optimization formulation). Following this procedure, we obtain an RA profile for each sub-region, which is the hourly loss of load probability (LOLP) time series. This RA profile contains the fraction of Monte Carlo iterations which resulted in a loss of load event in each hour. We refer to any hour with a $LOLP > 0$ to be a risk hour. As we find the LOLP time series, we also simultaneously calculate the expected hourly shortfall time series and the total expected unserved energy (EUE). The expected hourly shortfall is the sum of (load − generation) for those trials when load exceeds generation, divided by the total number of trials. EUE is the sum of this hourly expected shortfall.

Unlike our RAM, our CEM does not account for stochastic outages. Instead, the CEM aims to produce a resource adequate system by enforcing a planning reserve margin. To facilitate resource adequacy comparisons across future systems output by our CEM, our RAM adjusts the generation fleets in CAMX for each case we model so that each fleet's annual resource adequacy achieves a target value. Specifically, the RAM iteratively adds or removes NGCC capacity in CAMX then calculates annual resource adequacy until the annual loss of load hours ($LOLH = \sum(LOLP)$) is 2.4 in each case. This target value reflects the real-world 1-in-10 reliability standard widely adopted by utilities. Due to high computational time taken to obtain the RA profiles and apriori unknown number of addition/removal trials of NGCC capacity, the iterative procedure is performed with 50 Monte Carlo samples at each stage. This means that the final fleets all do not have an exact LOLH = 2.4, but vary between LOLH = 2 to LOLH = 2.6. After each generator fleet is adjusted, the RAM estimates the fleet's hourly and annual resource adequacy. We use CAMX as the sub-region of interest as it shows highest LOLH across the scenarios modeled and the timing of RA failure in other regions coincide with RA failures in CAMX.

Inputs to the RAM include the generator fleets output by the CEM; hourly surface air temperatures; and forced outage rates. The CEM provides location and sub-region specific installed capacities for all generators and storage. The CEM has various generators, but in going from CEM to RAM we retain these generators as such, but combine— pumped hydro, batteries, fuel cell to *storage* type; and geothermal, different types of waste, biomass, and other small fossil generators *other* type.

Prior to the stochastic simulation procedure, we calculate the hydroelectric generation for each scenario within each sub-region. For each of our five regions in WECC, we obtain monthly hydropower generation from EIA-923 data, then calculate sub-regional contribution proportional to installed capacity. To estimate hourly generation, we then carry out a greedy dispatch procedure for each month. The algorithm first quantifies hourly electricity demand not met by every generator other than hydropower and storage units (i.e., residual demand). The algorithm then dispatches hydropower units on a consecutive hourly basis. In each hour, the algorithm sets regional hydropower generation equal to the minimum of residual demand and regional total installed hydropower capacity, provided cumulative monthly generation through each hour doesn't exceed monthly generation limits. Any leftover monthly generation in the month is redistributed to all hours proportional to electricity demand minus wind and solar generation (i.e., net demand).

## Meteorological analysis

**Weather regimes.** To characterize meteorological drivers of risk hours, we begin by identifying the weather regimes and corresponding circulation patterns that coincide with risk hours. To identify weather regimes in our study region (WECC), we use self-organizing maps (SOMs), which is an unsupervised neural-network-based clustering technique. Unlike other hierarchical and non-hierarchical clustering techniques, SOMs cluster input data into nodes that form a topological representation in which node proximity indicates their similarity. Previous studies have identified weather regimes with SOMs in other contexts, e.g., to quantify the frequency and persistence of weather regimes associated with heat waves[56] and extreme precipitation events[57] in a warming climate.

We create our SOMs using seasonal anomalies of the daily average 500 hPa geopotential height (Z500) for the extended summer season (June through September, or JJAS) from 1981–2020. We analyze an extended summer season because our risk hours occur in June through September, so we focus on the warmest months of the year without narrowly constraining our SOMs to a small subset of months. We use Z500 because it captures synoptic-scale atmospheric processes and their relationship with surface meteorology, is persistent over multiple days, and is widely used for weather typing in the US and Europe[25,32,58,59]. To produce the SOM, we use the MiniSom Python package[60] with the following parameterization: grid shape of 3 rows and 3 columns, a *gaussian* neighborhood function, sigma (i.e., spread of neighborhood function) value of 1, learning rate of 0.1, and 5000 training iterations. These parameter values provide a concise weather regime representation that balances quantization and topographic error (see SI Section 4). SI Fig. A.6 shows the total number of days attributed to each weather pattern over the 40 year period used to train the SOM. Since the objective of weather patterning is not to get an equal number of elements in each node, but to cluster weather patterns based on similarity, the number of days assigned to all weather patterns are not equal.

**Surface meteorology.** While daily Z500 anomalies are a meaningful variable for weather regime identification via SOMs, the power system is directly affected not by Z500 but rather by surface meteorological variables. Thus, we study surface meteorology corresponding to the weather regimes as well as surface meteorology on the RA failure days for the different years. For each weather regime identified by our SOM, we make composite maps of hourly anomalies in surface temperature, surface solar radiation, and near surface wind speed. To calculate these hourly anomalies, we calculate the JJAS seasonal hour-of-day mean of surface weather data for each year (yielding 24 mean values for each year), then subtract this seasonal hour-of-day mean from each hourly data point within the years. We analyze anomalies within the year rather over the 40-year period as our models work with a yearly time series and that the investment decisions are made to cater to that year. Using the hourly anomalies, we construct composite maps for the weather years (2016–2019) in a two step process. First, we map each day from the extended summer months to a weather regime by

passing daily Z500 anomaly into the SOM. Second, for every hour of each day that belong to each weather regime, we average the hourly surface meteorology anomalies to get the composite surface meteorological anomalies under each weather regime. For solar radiation anomaly composites, we choose only the daylight hours region wide (6 a.m. to 8 p.m. PST) to avoid biasing the composites toward the hours with very low solar radiation. To capture surface meteorology directly driving the RA failure days, we find the unique days when these events occur across the four weather years analyzed at RE penetrations of 30% or more, and plot the mean surface meteorology anomaly in those days. Here too, for solar radiation anomalies we use only the daylight hours.

### Data description

**Demand data.** We get hourly sub-regional electricity demand from a database of screened and imputed data based on observed demand[44]. Due to limited availability of observed hourly electricity demand, the database provides four full years of balancing authority (BA) level demand from 2016 through 2019, and sub-regional demand is constructed by aggregating demand from BAs within each sub-region (ref. SI Section 2.6.1). Though there are techniques to backcast electricity demand based on meteorological and societal factors, these methods exhibit large errors, particularly in predicting extreme demand values[7,61]. Since demand extremes are a major factor in RA, we opt for observational rather than backcasted demand values.

**ERA5 reanalysis data.** Given that identification of weather regimes requires long-term (multi-decadal) weather data, we use reanalysis weather data for our analysis. Specifically, we obtain weather data from the ERA5 reanalysis dataset[62]. The weather data used for surface meteorological anomalies and weather pattern identification for each weather year coincides with the weather data used to drive the power system models for the corresponding weather year. We choose ERA5 because it provides wind speeds at 100 m above surface at hourly resolution, unlike other reanalyses products[63]. ERA5 is also widely used in power systems and synoptic meteorology research[24,34,35]. From ERA5, we specifically obtain near-surface air temperature (t2m); dew-point temperature (tdps); air pressure (sp); zonal and meridional surface wind speeds (u10 and v10); downward shortwave solar radiation at the surface (ssrd); and zonal and meridional wind speeds at 100 m level (u100 and v100). We obtain each data field at hourly temporal resolution and 30 km spatial resolution.

**Capacity factors.** We derive solar capacity factors directly from the surface downwelling shortwave radiation data for a EFG-Polycrystalline silicon photovoltaic module using the formulation described by Jerez et al.[64] (see SI Section 1.1). We calculate wind capacity factors using the formulation described by Karnauskas et al.[65] and the composite 1.5 MW IEC class III turbine from the System Advisor Model[66] (see SI Section 1.2).

**Technology and costs.** We obtain operational costs for existing generators from the NREL Annual Technology Baseline (ATB) moderate technology development scenario for 2030[67], and fuel costs from the EIA annual energy outlook for 2020[68]. For new units which the CEM determines investment in, we obtain capital costs from the ATB.

### Scenarios

To capture the effect of increasing renewable penetrations on meteorological drivers of reliability, we run four scenarios of increasing wind plus solar penetrations: 9 (based on the current fleet), 30, 45, and 60%. These scenarios are enforced in the CEM by constraining constraining annual wind plus solar generation to equal to a percentage of annual electricity demand. Given significant inter-annual variability in meteorology and climate, we run our modeling

framework for each renewable scenario for each year of available electricity demand data (2016 through 2019). This approach treats each meteorological year as an independent observation, allowing us to quantify the robustness of our results to different weather years.

While our results are based on fleets built for specified renewable penetrations, we have also explored publicly available datasets for understanding the plausibility of the fleets we have obtained. One of these, the WECC anchor dataset (ADS), provides generator fleet and hourly load and renewable generation shapes for 2032. The ADS renewable penetration percent is 32% with total installed capacity of 60GW in utility scale solar PV and 38GW of on-shore wind generation, which falls within our renewable penetration and installed generation ranges studied. While our methods can also be applied to that dataset to understand the meteorological drivers, we have not done so in this paper for conciseness.

## Data availability
Meteorological, power system output from the models, and code used to create the final figures in the manuscript are available via Zenodo[69]. Source data are provided with this paper.

## Code availability
Code for the CEM and RAM used in this study is available online via Zenodo[70].

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

## Acknowledgements

This work was supported by the National Science Foundation under the Award #2142421. S.S. thanks Dr. An Pham for her help with the CEM formulation & code and Isaac Bromley-Dulfano for help with the RAM. The results contain modified Copernicus Climate Change Service information 2020. Neither the European Commission nor ECMWF is responsible for any use that may be made of the Copernicus information or data it contains.

## Author contributions

S.S.—conceptualized and designed research, analyzed data, developed analytical pipeline, wrote paper; M.T.C.—conceptualized, designed and supervised research, wrote paper, provided funding; A.E.P.—conceptualized, designed, and supervised research, revised paper; F.L. and D.J.B.—supervised research, revised paper.

## Competing interests

The authors declare no competing interests.
