## [Peer Review File · Nature Communications]

Meteorological Drivers of Resource Adequacy Failures in Current and High Renewable Western U.S. Power SystemsReviewers' Comments:

Reviewer #1:

Remarks to the Author:

The high-level takeaway from this work seems to be that periods with high temperatures, low solar irradiance, and low wind are the greatest risk to the Western Interconnection (WI) - unfortunately, as a headline this is not a particularly surprising/novel insight for power system planners. It may be more useful to further emphasize and detail the common link between these conditions and high-pressure anomalies, although reading as a power systems practitioner, I would appreciate more explicit framing of how knowledge of this risky weather pattern can help improve planning practices.

This reviewer does not have the background to assess whether the link between these problematic system conditions and this weather pattern is a novel insight or not, but if so, it's not clear why it requires capacity expansion modelling or detailed resource adequacy assessment to study - neither provide significant additional insight into the fact that hot, cloudy, still periods present notable adequacy risk to the power system. Reduced emphasis on these grid modeling aspects in favor of a deeper dive into the underlying meteorological phenomena may provide more overall value.

Taking the broader premise of the paper and grid modeling focus at face value, the lack of transmission constraints in both the capacity expansion model and resource adequacy model is, in my view, problematic. Both least-cost generation portfolios and shortfall risk conditions will look very different depending on the presence or absence of transmission constraints, which undermines the perceived applicability of these results. Many of the questionable quirks in the capacity modeling results (e.g. little-to-no perceived value of wind or storage) likely stem from over-optimistic assumptions about power delivery across the WI.

More broadly, I think additional justification is needed to motivate developing original future generating portfolios using a capacity expansion model. The specific nature of weather sensitivities in a future system are presumably highly dependent on the specifics of the future portfolio, which could be sensitive to a number of factors not considered here. There are many more thoroughly vetted datasets, such as the WECC 2032 Anchor Dataset or NREL's Standard Scenario suite, that provide plausible future system conditions with more detailed considerations of load growth scenarios, transmission constraints, state and federal policy, announced generator builds and retirements, etc. It's not clear why these datasets wouldn't have been sufficient (or even preferable) for this work.

From an adequacy assessment perspective, while a full nodal transmission representation may be computationally impractical, I'm not aware of any generally-accepted adequacy assessment framework for the full Western Interconnection that doesn't at least consider regional interface constraints between sub-regions or balancing authorities. Different regions of the WI have different seasonal risk profiles (with the Pacific Northwest presently having more winter risk while the rest of the system has more summer risk), and neglecting interregional transmission constraints washes out these spatial dynamics, which are presumably important to consider when assessing weather patterns that drive adequacy risk. If it's felt that these spatial effects are less significant than conventionally assumed, an empirical justification to support this position would seem necessary to convince readers of the relevance of these results to the real WI power system.

Regarding the resource adequacy analysis and classification of weather patterns, one key discussion that seems to be missing concerns the number of hours assigned to each weather regime - how do we know that weather patterns associated with high-pressure anomalies aren't simply more common than low-pressure anomalies, and as such make up a larger fraction of the hours of the year (thus increasing the prior probability that a risk period occurs in a high-pressure anomaly period)? Information about the count of expected event-hours (LOLH) relative to the number of hours assigned to each weather pattern seems important to include.

Furthermore, limiting the weather pattern clustering to summer months neglects risk periods for the Pacific Northwest, which has higher winter loads - although as discussed above, this dynamic would not be discernable to the grid models in the absence of transmission constraints anyways. The more problematic implication of this decision may be that, if heating electrification trends evolve in line with existing state decarbonization policies, future adequacy risk could be expected to shift towards the winter, lessening the importance of summer high-pressure anomalies to system adequacy and undermining the findings of this work. This points to a broader challenge with this work, which is that the weather patterns driving adequacy risk are dependent on the timing of shortfall risk, which in turn is subject to many uncertainties and factors not considered in the work. (If it could be shown that, under a much wider range of potential system futures than considered here, the weather patterns driving system risk remain consistent, that would be an interesting finding).

The phenomenon of increasing concentration of risk into less frequent but higher-likelihood shortfall periods is interesting and seems to merit further discussion, with care being taken to disentangle the extent to which this is simply a modeling artefact (given the fact that LOLH is held constant across systems, such that hourly outage risk and frequency of risk periods must by definition be inversely proportional) as opposed to a more fundamental characteristic of systems with higher levels of wind and solar. Careful consideration should also likely be given to the fact that these systems are being designed and tested against (from what I understand) identical weather years, such that the planning model has perfect foresight into wind and solar variability, and the only uncertainty being assessed by the RA model stems from thermal generator outages. This is obviously flipped from actual practise where, on planning timescales, the uncertainty around wind and solar availability in any hour is likely to exceed that associated with thermal units.

Finally, it wasn't clear to me whether storage was actually included in the analysis? The capacity expansion discussion seems to imply the only result that yielded a storage buildout was re-run with the storage option removed, but the adequacy assessment discussion implied that storage was re-dispatched for each of the 10,000 outage draws (however, if this was indeed the case, it's unclear why hydropower wasn't dispatched similarly). A clearer explanation of the methods used here would be appreciated.

A few other smaller suggestions for improvement:

- LOLE/LOLH is increasingly being recognised as a problematic primary risk metric given its inability to reflect the magnitude of shortfall events. It may be worth reporting adequacy results (and/or tuning capacity expansion levels) based on expected unserved energy (EUE) as well as or instead of LOLH.
- The introduction implies equivalence between the recent California and Texas events and blackouts, which is not strictly accurate in the technical sense of the term: both of those events were characterized by rolling brownouts, controlled involuntary load shedding intended to avoid destabilizing frequency drops and protect the system from a full-scale grid collapse, or "blackout", which would be substantially more widespread as well as slower and more involved to recover from.
- The "star" plots communicating the temporal distribution of shortfall risk in Figures 1 and 2 seem less-than-ideal given the density of markers and the fact that individual markers obscure one another. The standard means of communicating these distributions of temporal risk in adequacy studies is to use a 2D heat map with the same axes, but where each hour is associated with a non-overlapping rectangle and the color of the rectangle indicates shortfall risk magnitude (e.g. ranging from white, no risk, to dark red, high risk).

Reviewer #2:
Remarks to the Author:

This paper addresses an important topic area on exploring the meteorological drivers for power system resource adequacy (RA) performance. The authors map the system RA performance with different meteorological patterns for the western United States, and analyze the linkage of the two under alternative variable renewable penetration levels. Overall, the paper provides novel insights into an important research question. The manuscript is well-written and the modeling methods are clear.

Some suggested changes and concerns are summarized below:

1. Definition and the use of "RA": the term "RA" in this paper is vaguely defined and can result in confusions. While the definition is cited from a NERC report, "RA" is generally used to indicate long-term planning-scale system reliability (as opposed to short-term operational reliability). Even though RA is linked more closely with operational aspects recently, the authors should clarify the definition and the use of RA they are discussing in this paper.

Also, the authors should clarify what aspects of "RA" they are referring to when mentioning "RA". For example, on page 3 line 118, the research question part: circulation patterns do not drive "RA", they can impact RA performance. Also, on page 3 line 134, "RA profile" is not a standard term, please consider adding further explanations to it or replace the term. Please check throughout the paper to make all the uses of "RA" terms more explicit.

2. Contribution and application of the work: this paper seems to analyze how meteorological drivers impact "RA performance" given fixed resource mix, and does not consider how these RA performance issues can reversely impact investments, i.e., this is a one-way impact analysis. The work highlights the needs to incorporate these meteorological and operational impacts in the RA assessments to incentivize an adequate system supply, because essentially these assessments and the suggested higher risks should all be linked back to adjusting system investment. It also suggests purely based a stylized planning reserve margin value in planning and investment decisions are not sufficient to guarantee operational reliability. The work can also provide insights to short-term operation by mapping certain weather patterns with reliability risks, and therefore suggesting red flags in system operation when certain patterns show up in forecasts.

As such, the authors should 1) when mention "RA", clarify that these are performance or levels or something else. For example, on page 2 line 85 (and other places), "RA failures" indicates "performance failure" rather than "investment failure" because the system already meets the identified planning reserve margin in the capacity expansion model; 2) modify the Introduction section to make the above point (one-way impact analysis) and the contribution more clear, and rephrase key sentences such as (among others) "this paper aims to better understand the meteorological drivers of RA and how investment decisions in renewable energy affect those drivers" (page 2, line 70) – meteorological driver of RA performance, and investment decisions should not be a static portfolio to affect drivers.

3. Better explanation of methods and assumptions justifications:

- Why choosing 0.005 as the threshold for "risk hours"?
- Examples of 12 weather regimes to illustrate what types of key weather phenomenon they represent (e.g., a major heat/cold wave event etc.), especially for WP 8, 9 and 10 ones due to their significant impacts.
- Relationships between Z500 anomalies data, surface meteorology data and temperature data used for RAM modeling: for example, are the temperature data for surface meteorology the same as the temperature used in RAM for temperature-dependent outage calculation?

4. Page 13, line 565-574, the authors discussed the generator fleet adjustments to match the LOLH=2.4 requirement, such that the system has equivalent RA performance with the capacity expansion results. Why is this necessary? The planning reserve margin of 15% in the CEM is an approximation itself, and it is not necessarily equivalent to a LOLH=2.4 system. The LOLH value of the system mix from the CEM should be what comes from the RAM simulations. Manually adjusting the

generators can create discrepancies between the two systems.

5. The RAM analysis is an 8760 hourly simple Monte Carlo simulation, whereas in the real world the system operation can have multi-day-ahead to day-ahead or intra-day forecasts and operational adjustments if the operator foresees the weather events. Please add discussion on how ignoring these impacts may impact your results.

6. Other minor questions/comments:

- Page 2, line 80, this is the first time the term "GCM" occur. Please add it, otherwise there is no reference of "GCM" term for following usage
- Figure 1 and 2: the figures are too small

Dear Reviewers,

Thank you for the opportunity to revise our manuscript, and for the excellent comments we received. In response to reviewer comments, we have made substantial changes to our manuscript, including significant improvements to our methods, additional results, and additional discussion. We think these changes are fully responsive to the reviewer's concerns, and have significantly improved our paper's quality and impact. While we have detailed responses to comments below, we want to flag two major changes relevant to all reviewers here.

Major Change 1: Method updates:

We now include transmission between sub-regions of the western interconnection (WECC) in capacity expansion model (CEM) and resource adequacy model (RAM) (*See response to comment 1.3 and SI section 2,3 for more details*).

Major Change 2: Change in analysis scope to California subregion of WECC :

With the inclusion of transmission, we find that risk (and timing of risk) is predominantly driven by the CAMX region within WECC, so we shift our analysis scope to this region (*See response to comments 1.3 and 1.5*).

Despite these major changes, our key results have not qualitatively changed, illustrating the robustness of our analytical framework and approach.

Reviewer #1 (Remarks to the Author):

1.1 The high-level takeaway from this work seems to be that periods with high temperatures, low solar irradiance, and low wind are the greatest risk to the Western Interconnection (WI) - unfortunately, as a headline this is not a particularly surprising/novel insight for power system planners. It may be more useful to further emphasize and detail the common link between these conditions and high-pressure anomalies, although reading as a power systems practitioner, I would appreciate more explicit framing of how knowledge of this risky weather pattern can help improve planning practices.

Thank you for the suggestions. We would instead frame the high-level takeaway of our paper as:

“Weather patterns representing upper atmospheric high pressure anomalies drive resource adequacy failures in the Western Interconnection under current and high levels of renewables penetrations.” While we agree with the reviewer’s interpretation of the connection between each component of surface meteorology to system risk, we believe the weather pattern approach can add additional value. We also agree with the reviewer that more explicit framing of how weather patterns can improve planning practices would be valuable, and underscore the contribution of our work. To that end, we have added text to the introduction and discussion sections as follows:

Starting at line 124 in Introduction:

“The spatial coverage of these large-scale atmospheric circulation patterns makes them valuable analogues for surface meteorology over large geographic regions. Using these synoptic drivers in planning and operations can benefit system operators when thinking about RA due to current and future systems' increasing dependence on generation over larger areas and interconnected balancing authorities.”

Starting at line 451 in Discussion:

“The added value that our weather pattern approach gives over just a surface meteorological analysis is that we are able to capture the synoptic scale (1000-2500km) drivers of the RA failure events. The weather patterns can be used in different ways to incorporate meteorological drivers of the power system in system planning as well as operations as we move to interconnected continental scale systems. For system planning purposes, current practices mostly involve only using historical meteorological data with techniques like importance subsampling reducing computational costs by providing representative periods to the capacity expansion model. Our findings can improve this subsampling process by providing a physical basis of choosing the representative periods. Further, to make informed investment decisions and maintain system reliability in the future, system planning needs to use future meteorological data from climate projections and the physics based subsampling procedure can help here as well. Future climate projections from global climate models have lower spatial and temporal resolution than required by power system models. Incorporating this future climate data requires computationally costly downscaling. Our methods can reduce downscaling needs and associated costs by guiding

selective downscaling of certain time periods of interest, e.g. time periods with high pressure anomalies in the Western US, to drive system planning and operation models. This can help system planners understand further risks, beyond resource adequacy, during these stressful periods. At the operational level, system operators, utilities, power producers, and communities can use the short term forecasts at the days to weeks timescale and long-range probabilistic forecasting at season-to-season time scale to avoid scheduling maintenance and other related down times when these patterns are expected to occur. These patterns are characterized by their temporal persistence and ability to represent meteorology at the synoptic scale during the occurrence of extreme events. These characteristics make the WPs more suitable, as an aggregate pointer to capture stressful periods for system operations, than individual surface meteorological variables, which exhibit higher spatiotemporal variations. ”

The reviewer also suggests to better link surface weather conditions with weather patterns, including high-pressure anomalies. In our original manuscript, we link each weather pattern to composites of surface temperature, wind speed, and solar radiation. These composites indicate average anomalies for all days that belong to each weather pattern. We have extended on this link to include the dynamic mechanisms which drive these high pressure patterns to cause the corresponding surface meteorology in our introduction as (bold indicates new additions):

“The resulting large-scale patterns have strong associations with surface-level meteorological variables that directly affect the power system, including extreme surface air temperatures. **These patterns indicate several processes like temperature advection and subsidence which can, under certain conditions, drive extreme events in the power system.**”

1.2 This reviewer does not have the background to assess whether the link between these problematic system conditions and this weather pattern is a novel insight or not, but if so, it's not clear why it requires capacity expansion modelling or detailed resource adequacy assessment to study - neither provide significant additional insight into the fact that hot, cloudy, still periods present notable adequacy risk to the power system. Reduced emphasis on these grid modeling aspects in favor of a deeper dive into the underlying meteorological phenomena may provide more overall value.

First, while we appreciate the reviewer's candor, we do want to make clear that our research is the first to link power system outcomes to large-scale weather patterns in a United States context, and the first to link power system resource adequacy to weather patterns. As discussed in our response to the prior comment, we think this linkage provides significant value for several reasons. We make this novelty clear in the following text from our manuscript:

“Our research is the first to link weather patterns and power systems operations in the United States, and the first to characterize weather regimes driving RA failures.”

Second, our power system models are designed to answer the specific question of the interaction between meteorology, renewable energy penetrations, investment decisions, and the consequent resource adequacy failures. To examine fleets which are built with higher renewable energy (RE) penetrations, we need a capacity expansion model. In order to analyze the resource adequacy (RA) failures and examine weather patterns which lead to these failures, we need a resource adequacy model. Thus, our modeling framework couples a capacity expansion and resource adequacy model, allowing us to study failures in future power systems with increasing renewable penetrations. We explain this modeling framework in the following text from our manuscript:

“We first construct fleets that generate increasing levels of wind and solar electricity (hereafter “renewable electricity” or RE) using a capacity expansion model (CEM) (see Methods.3.2). The CEM is a deterministic linear program that minimizes fixed plus variable costs of electricity generation, storage, and transmission investments and operations. The CEM specifically optimizes investments in wind, solar, 4-hour electricity storage facilities, and inter-regional transmission capacities, and optimizes operations of existing and new generation, storage, and transmission. The CEM does not optimize investment in new thermal facilities given its coupling with our RAM, which adds or removes thermal facilities to reach a given reliability target. Investment and operational decisions are subject to numerous generator- and system-level constraints, including hourly balance of supply and demand. To capture co-variability and extremes in electricity demand and wind and solar generation, we use observed hourly electricity demand for WECC and coincident spatially-differentiated RE capacity factors (see Methods.3.5.3). In our models we resolve WECC into five constituent sub-regions, as used by WECC in its Western Assessment of Resource Adequacy report (ref SI fig. A.3). Between each pair of sub-regions, we model transmission flows using the transport method. Investment decisions in storage occur at the five-region level; in transmission between each pair of regions; and in wind and solar at spatially-differentiated resource locations on a roughly 30 by 30 km grid. RE penetration levels are enforced at the WECC scale.

We then quantify a RA profile for each fleet and each sub-region from the CEM using a resource adequacy model (RAM), which simulates stochastic forced outages of generators using a non-sequential Monte Carlo sampling procedure and finds hours where there is a non-zero probability of demand exceeding total available generation (see Methods.3.3). We use empirically-derived temperature-dependent forced outage rates for NGCC and hydropower facilities and constant outage rates for other generators. From the RAM, we obtain a timeseries of loss of load probabilities (LOLPs) by hour of the year, which we refer to as the RA profile. This RA profile is a function of short-term operations from the RAM. Hours with LOLPs greater than zero indicate a risk of an RA failure; we refer to these hours as “RA risk hours” or “risk hours”

Third, the reviewer questions what additional insights our analysis has yielded. We partly answered this question by our prior response (see 1.1) and related text additions to the manuscript. To reiterate, there is significant value in understanding large-scale weather patterns

that drive surface-level conditions that contribute to failures for planning and forecasting purposes in the mid- and long-run, especially as Western power systems become increasingly interconnected. Our new results (as explained above with respect to surface meteorology on the RA failure days) also illustrate a diversity of surface conditions that can lead to failures.

Finally, the reviewer suggests “[r]educed emphasis on these grid modeling aspects in favor of a deeper dive into the underlying meteorological phenomena...” We have also explored surface meteorology on each day in which RA failures occur.

Starting at line 532 in Results

“At higher RE penetrations, the risk is attributed to fewer days. So we look at the daily average temperature anomalies for these days [Figure 7]. Though these days are driven by WPs 6,7, or 8 across the weather years, they represent different distribution of surface meteorological anomalies in the different years. On the RA failure days, the temperature anomalies across these four years show predominantly positive anomalies over large portions of the region, but the magnitude, geographical location and extent of the positive anomalies vary. Some days also exhibit negative anomalies in some regions, but even on these days the anomalies are positive in the California region. SI figs. A.14 and A.15 show the surface solar radiation and wind speed anomalies for these days.”

Details about the methods for this change (line 872):

“To capture surface meteorology directly driving the RA failure days, we find the unique days when these events occur across the four weather years analysed at RE penetrations of 30% or more, and plot the mean surface meteorology anomaly in those days. Here too, for solar radiation anomalies we use only the daylight hours.”

1.3 Taking the broader premise of the paper and grid modeling focus at face value, the lack of transmission constraints in both the capacity expansion model and resource adequacy model is, in my view, problematic. Both least-cost generation portfolios and shortfall risk conditions will look very different depending on the presence or absence of transmission constraints, which undermines the perceived applicability of these results. Many of the questionable quirks in the capacity modeling results (e.g. little-to-no perceived value of wind or storage) likely stem from over-optimistic assumptions about power delivery across the WI.

We appreciate the suggestion to include transmission constraints in our resource adequacy model. We have made significant enhancements to our capacity expansion and resource adequacy models to address these concerns by the reviewer, including by adding transmission to both models. To align our analysis with WECC reliability assessments, we use the same regional boundaries as the WECC Western Assessment of Resource Adequacy (see Figure 1 below). We detail model enhancements below. Despite significant modeling enhancements, our results do not qualitatively change: across all studied renewable penetrations and weather years, high pressure weather patterns centered over the western US drive surface conditions which cause reliability failures.

Figure 1: WECC subregions used in the CEM and RAM

In adding transmission constraints, reliability failures occur at the sub-regional level, not at the WECC level. Thus, we refocus our analysis from WECC reliability failures to CAMX (specifically the US portion of CAMX) reliability failures. We focus on CAMX in our analysis for several reasons, as explained in our revised manuscript:

“We restrict our analysis to the CAMX region for two reasons. First, NERC's Long-Term Reliability Assessment (LTRA) indicates CAMX is the most vulnerable WECC region to resource adequacy failures in the near term, with LOLH of 0.72 and 9.79 in 2024 and 2026 respectively in the 2022 assessment. By comparison, other regions in WECC have LOLH of up to 0.03 (2024) and 0.37 (2026), an order of magnitude less than CAMX. Thus, understanding meteorological drivers of RA failures in CAMX can provide significant near-term value to decision makers and serve as a model for analyses in future regions. Our resource adequacy results agree with the LTRA, as we find CAMX has at least 4x and 27x more probability of resource adequacy failure than any other WECC region in the current and RE penetration greater than 30% fleets respectively across the years. Second, we find that in all but one scenario we analyze, and in all RE penetration greater than or equal to 30%, the CAMX risk hours coincide with risk hours in other regions if failures occur in other regions.”

We have made several major modifications to our methods to better capture transmission and other system factors. First, we have modified our methods to now include transmission in both the capacity expansion model and the resource adequacy model. The transmission representation is at the WECC subregional level within the United States, as modeled by WECC in their Western Assessment of Resource Adequacy. The model details contained in section 3.2 (CEM) and 3.3 (RAM) discuss the ways in which this inter-regional transmission is treated in the respective models.

Second, we have modified how we dispatch hydropower in the resource adequacy model. This is also explained in detail in section 3.3 (RAM):

“Prior to the stochastic simulation procedure, we calculate the hydroelectric generation for each scenario within each sub-region. For each of our five regions in WECC, we obtain monthly hydropower generation from EIA-923 data, then calculate subregional contribution proportional to installed capacity. To estimate hourly generation, we then carry out a greedy dispatch procedure for each month. The algorithm first quantifies hourly electricity demand not met by every generator other than hydropower and storage units (i.e., residual demand). The algorithm then dispatches hydropower units on a consecutive hourly basis. In each hour, the algorithm sets regional hydropower generation equal to the minimum of residual demand and regional total installed hydropower capacity, provided cumulative monthly generation through each hour doesn’t exceed monthly generation limits. Any leftover monthly generation in the month is redistributed to all hours proportional to electricity demand minus wind and solar generation (i.e., net demand).”

Third, we have overhauled our capacity expansion modeling approach to align with other state-of-the-art models. Our expansion model now captures transmission, invests in new renewables at locations with and without existing renewables, and runs brownfield beginning with the existing WECC fleet. For a full description of our updated CE model, see SI section 2.

1.4 More broadly, I think additional justification is needed to motivate developing original future generating portfolios using a capacity expansion model. The specific nature of weather sensitivities in a future system are presumably highly dependent on the specifics of the future portfolio, which could be sensitive to a number of factors not considered here. There are many more thoroughly vetted datasets, such as the WECC 2032 Anchor Dataset or NREL’s Standard Scenario suite, that provide plausible future system conditions with more detailed considerations of load growth scenarios, transmission constraints, state and federal policy, announced generator builds and retirements, etc. It’s not clear why these datasets wouldn’t have been sufficient (or even preferable) for this work.

We appreciate the suggestion to refocus on other available datasets. Two are suggested here: the 2032 Anchor Dataset and NREL’s Standard Scenario suite. Neither has adequate publicly-available information for our analysis. NREL’s Standard Scenario suite lacks location-specific data for renewable resource deployments, which is essential to use different weather years for wind and solar generation in our analysis. The 2032 Anchor Dataset lacks transmission data, which is necessary to model regional resource adequacy per the reviewer’s suggestions. Furthermore, a key motivating factor of our study is to understand how drivers of resource adequacy failures might evolve with increasing renewable penetrations. The Anchor Dataset does not contain different renewable penetration levels. However, the ADS renewable penetration percent is 32% with total installed capacity of 60GW in utility scale solar PV and 38GW of on-shore wind generation, which falls within our renewable penetration and installed

generation ranges studied. We have added a reference to the Anchor Dataset and the value of our results in reference to that dataset in the scenarios section of methods (Section 3.6):

“While our results are based on fleets built for specified renewable penetrations, we have also explored publicly available datasets for understanding the plausibility of the fleets we have obtained. One of these, the WECC anchor dataset (ADS), provides generator fleet and hourly load and renewable generation shapes for 2032. The ADS renewable penetration percent is 32% with total installed capacity of 60GW in utility scale solar PV and 38GW of on-shore wind generation, which falls within our renewable penetration and installed generation ranges studied. While our methods can also be applied to that dataset to understand the meteorological drivers, we have not done so here for conciseness.”

Additionally, our experimental design generates 12 different future fleets: a fleet for all combinations of 4 weather years and 3 renewable penetrations. Thus, while none of these fleets perfectly predicts the future WECC fleet, they collectively test a wide range of possible future fleet realizations, adding robustness to our results.

1.5 From an adequacy assessment perspective, while a full nodal transmission representation may be computationally impractical, I’m not aware of any generally-accepted adequacy assessment framework for the full Western Interconnection that doesn’t at least consider regional interface constraints between sub-regions or balancing authorities. Different regions of the WI have different seasonal risk profiles (with the Pacific Northwest presently having more winter risk while the rest of the system has more summer risk), and neglecting interregional transmission constraints washes out these spatial dynamics, which are presumably important to consider when assessing weather patterns that drive adequacy risk. If it’s felt that these spatial effects are less significant than conventionally assumed, an empirical justification to support this position would seem necessary to convince readers of the relevance of these results to the real WI power system.

We appreciate the reviewer highlighting these aspects of WECC reliability. As noted above, we have overhauled our modeling framework to capture transmission constraints in our planning and resource adequacy models, partly allaying the concerns stated here. The reviewer points to the Pacific Northwest as possibly presenting a different risk profile than other regions. But, from our analysis we find that if outages occur in regions other than CAMX, in 15 of the 16 scenarios modeled, these outages coincide with the CAMX outages, hence the western interconnection as a whole experiences the same weather pattern. Despite the change in the model formulation, where we have included a representation of transmission at the sub-regional scale, we still do find the risk hours occur in the summer, and occur only in the CAMX region in 12 of the 16 scenarios. In the other cases CAMX risk far outweighs the risk in other regions. This has been further detailed above in response to the reviewer’s third comment.

1.6 Regarding the resource adequacy analysis and classification of weather patterns, one key discussion that seems to be missing concerns the number of hours assigned to each weather

regime – how do we know that weather patterns associated with high-pressure anomalies aren't simply more common than low-pressure anomalies, and as such make up a larger fraction of the hours of the year (thus increasing the prior probability that a risk period occurs in a high-pressure anomaly period)? Information about the count of expected event-hours (LOLH) relative to the number of hours assigned to each weather pattern seems important to include.

Thank you for suggesting we include this statistic, which we agree would help illuminate the unique role certain weather patterns have in driving RA failures. We have included a new figure in SI Figure 11 providing the number of days in the summer across our 40 years that belong to each pattern, reproduced below. In general, days per WP do not explain our results. WP 7, the dominant driver of RA failures, has a similar or lower number of days than other WPs (5, 6, and 8); and collectively, WPs 6, 7, and 8, which drive 100% of RA failures, account for only 38% of all extended summer days.

Figure 2: *Weather patterns representing the weather regimes with the titles for each panel indicating the number of extended summer days from June-September from 1981-2020 that fall into each weather regime*

In general, the objective of weather patterning is not to get an equal number of elements in each node, but to cluster weather patterns based on similarity. Furthermore, while the above figure shows frequency of days in each weather pattern across 40 years, we can also examine their frequency in our years of analysis relative to historic years (SI Figure 12, and reproduced below). Among our study years (2016 through 2019), 2 years have above trend line occurrences of WPs 7 and 8, and 2 years have below trend line occurrences of WP 7. This variability in WP frequency despite robustness of these WPs in driving RA failures between years gives more confidence in our results.

Figure 3: Grey dots show the percentage of extended summer days from 1981 - 2020 belonging to each weather regime. Red (negative slope) and blue (positive slope) dotted lines show a linear regression if the trend is greater than or equal to $|0.05|$ and bold parenthesized text indicates a 95% statistical significance of regression coefficient.

We have included these figures and explanatory text on this point to the manuscript as follows:

In results discussing the importance of WP 7:

“We find that the number of days attributed to the extreme weather patterns (WP 7 and WP 8, but particularly WP 7) are comparable to the number of days attributed to intermediate weather patterns (such as WPs 4, 5, and 6) [SI fig. A.6]. Moreover, among our study years, 2 years have above trend line occurrences of WPs 7 and 8, and 2 years have below trend line occurrences of WP 7. Despite the total number of days in each WP and variability in occurrence frequency among the years analyzed, WP7 emerges as the more dominant driver at higher RE penetrations across the weather years.”

In methods section 3.4.1:

“SI fig. A.6 shows the total number of days attributed to each weather pattern over the 40 year period used to train the SOM. Since the objective of weather patterning is not to get an

equal number of elements in each node, but to cluster weather patterns based on similarity, the number of days assigned to all weather patterns are not equal.”

1.7 Furthermore, limiting the weather pattern clustering to summer months neglects risk periods for the Pacific Northwest, which has higher winter loads - although as discussed above, this dynamic would not be discernible to the grid models in the absence of transmission constraints anyways. The more problematic implication of this decision may be that, if heating electrification trends evolve in line with existing state decarbonization policies, future adequacy risk could be expected to shift towards the winter, lessening the importance of summer high-pressure anomalies to system adequacy and undermining the findings of this work. This points to a broader challenge with this work, which is that the weather patterns driving adequacy risk are dependent on the timing of shortfall risk, which in turn is subject to many uncertainties and factors not considered in the work. (If it could be shown that, under a much wider range of potential system futures than considered here, the weather patterns driving system risk remain consistent, that would be an interesting finding).

We agree with the reviewer that interactions between decarbonization decisions and drivers of RA failures, both in terms of large-scale weather patterns and surface-level meteorology, is a crucial area of further inquiry. Adding the electrification angle, though, is out of scope for our work, and is an area we are actively pursuing. Electrification would require coupling our models with a bottom-up set of building models, a significant undertaking. We have noted this as a fruitful area for further inquiry in our Discussion:

“Second, future research could also incorporate decarbonization-driven changes on demand including electrification of residential heating and charging of electric vehicles. These extensions face several challenges, though, including estimating electricity demand with bottom-up models and obtaining high spatio-temporal resolution climate model outputs.”

1.8 The phenomenon of increasing concentration of risk into less frequent but higher-likelihood shortfall periods is interesting and seems to merit further discussion, with care being taken to disentangle the extent to which this is simply a modeling artefact (given the fact that LOLH is held constant across systems, such that hourly outage risk and frequency of risk periods must by definition be inversely proportional) as opposed to a more fundamental characteristic of systems with higher levels of wind and solar.

We have chosen not to focus on this concentration, as our research question is driven by the meteorological drivers of RA failures rather than the concentration of RA failures with increasing renewable penetrations. In general, though, as wind and solar penetrations reach high levels, we think it is a fundamental characteristic of those systems that risk will increasingly concentrate into fewer hours *if* system planners build systems to meet a reliability threshold, as we have done. At very high wind and solar penetrations, many hours of the year will have wind and solar

generation that make up a large percent or exceed total electricity demand. In a few hours of the year, wind and solar generation will cover a small portion of demand, which is when other generators must meet demand. It is in these hours when risk will accrue; as the number of these hours dwindle, risk will increasingly accrue to those hours.

1.9 Careful consideration should also likely be given to the fact that these systems are being designed and tested against (from what I understand) identical weather years, such that the planning model has perfect foresight into wind and solar variability, and the only uncertainty being assessed by the RA model stems from thermal generator outages. This is obviously flipped from actual practice where, on planning timescales, the uncertainty around wind and solar availability in any hour is likely to exceed that associated with thermal units.

The reviewer's interpretation is correct. We test four different weather years - 2016 through 2019 - with unique demand, weather, and wind and solar resource profiles. For each year, we use the year's weather to create a future generator fleet with the capacity expansion model, then quantify the resource adequacy of the resulting fleet with the same year's weather. We first checked the text to ensure this was clearly conveyed. We believe this text well describes this design (bold text indicates additions):

“By running this integrated modeling framework for four weather years (2016 through 2019) and RE penetrations (Current, 30%, 45%, and 60%, see sec. 3.6 for definition of RE penetration), we quantify the effect of increasing renewables on meteorological drivers of RA and the robustness of this effect across **independent** weather years. While using four weather years does not sample the full distribution of possible weather events and associated impacts on RA **and RA failures**, it does cover over 35,000 hours and permits us to use observed hourly electricity demand with coincidental wind and solar generation.”

The data from the four weather years include large variability in solar and wind resources. Our approach differs from many instances of power system planning in research and application, including in real-world planning exercises, that rely on a single weather year or even a subset of a single weather year. In leveraging interannual variability of meteorology, we are able to account for some level of uncertainty in wind and solar generation potential. How this affects the investment decision can be seen from the figure 4 below. For RE penetration levels of 60%, and for the different weather years, we see how variability influences the least cost power system in each of the sub-regions. Additionally, we see that there is also spatial heterogeneity in the investment variation which is illustrated by the capacity change in wind investment. For ex., higher wind investment in NWPP_Central does not imply higher wind investment in NWPP_NE. Despite heterogeneity in investment portfolios between weather years, our finding regarding weather patterns is robust across these years, as we note in our results:

“Meteorological drivers of RA failures are also robust to weather years [Figure 6]. WPs 6, 7, and 8, which are high pressure anomalies, drive most RA failures across all weather years. Collectively, these WPs drive 87% to 100% of all risk hours and 96% to 100% of cumulative LOLP across weather years.”

We would also note that we treat wind and solar resources deterministically, i.e. ignore uncertainty either in planning our resource adequacy, throughout our analysis. However, most, if not all, plans advanced by utilities similarly treat wind and solar deterministically. From a resource adequacy perspective, accounting for uncertainty in wind and solar resources would affect resource adequacy vis-a-vis thermal commitment decisions on a day-to-day basis, which is outside the scope of our work.

Figure 4: *Installed capacities of different generators for RE penetration of 60% across the 4 weather years.*

1.10 Finally, it wasn't clear to me whether storage was actually included in the analysis? The capacity expansion discussion seems to imply the only result that yielded a storage buildout was re-run with the storage option removed, but the adequacy assessment discussion implied that storage was re-dispatched for each of the 10,000 outage draws (however, if this was indeed the case, it's unclear why hydropower wasn't dispatched similarly). A clearer explanation of the methods used here would be appreciated.

In our updated methods, we include storage as an investment option in the CEM, storage operations of existing and new units in the CEM, and storage operations in the RAM. Since we do not have data about storage outages (forced outage rates), we do not sample for outages in

storage. But, the storage assets are independently dispatched for each Monte Carlo iteration. Storage parameters and constraints are detailed in SI section 2.4.3. In general, the CEM and RAM incorporate existing pumped hydropower storage and invest in new 4-hour lithium ion devices. Our CEM builds varying amounts of storage across weather years, from 1.9 to 13.8 GW, as detailed in the figure 5 below. This is besides the pumped hydro storage and other battery capacity existing in the current initial fleets.

Figure 5: Power capacity of new 4-hour battery storage systems built by the CEM across RE penetrations

1.11 A few other smaller suggestions for improvement:

1.11.1 - LOLE/LOLH is increasingly being recognised as a problematic primary risk metric given its inability to reflect the magnitude of shortfall events. It may be worth reporting adequacy results (and/or tuning capacity expansion levels) based on expected unserved energy (EUE) as well as or instead of LOLH.

Thank you for this suggestion. We have added a short analysis on how EUE changes across the different systems we study in the results section, and added the following figure 6 in the SI. The added results text is as follows:

Results line 458:

“For all the weather years and renewable penetrations, we also simultaneously calculate the effective unserved energy (EUE). This is the sum of expected shortfall (in GWh) during each risk hour. SI.fig. A.10 shows the EUE for the different systems with the effective shortfalls

ranging from 3.5 GWh to 4.6 GWh and 1.1 GWh to 3 GWh at 9% and 60% RE penetrations respectively.”

Figure 6: LOLH and EUE from CAMX for the different weather years and RE levels.

In section 3.3 (RAM methods) we have also added detail on how we calculate EUE:

“As we find the LOLP time series, we also simultaneously calculate the expected hourly shortfall time series and the total expected unserved energy (EUE). The expected hourly shortfall is the sum of (load - generation) for those trials when load exceeds generation, divided by the total number of trials. EUE is the sum of this hourly expected shortfall.”

Since we have not found any standard EUE metrics (like LOLH of 2.4) utilized in system planning, we have not tuned the fleets based on EUE.

1.11.2 - The introduction implies equivalence between the recent California and Texas events and blackouts, which is not strictly accurate in the technical sense of the term: both of those events were characterized by rolling brownouts, controlled involuntary load shedding intended to avoid destabilizing frequency drops and protect the system from a full-scale grid collapse, or “blackout”, which would be substantially more widespread as well as slower and more involved to recover from.

Thank you for noting this. We have modified the last line in introduction to:

“RA failures, ..., are often responsible for large-scale rolling outages, e.g. in California in 2020 and Texas in 2021. These two events were caused by a combination of higher than anticipated demand, due to a heatwave (in CA) and a cold snap (in TX), and generator outages driven by extreme weather. This necessitated intervention, like rolling outages, from the system operator to prevent catastrophic consequences to the system.”

1.11.3 - The “star” plots communicating the temporal distribution of shortfall risk in Figures 1 and 2 seem less-than-ideal given the density of markers and the fact that individual markers obscure one another. The standard means of communicating these distributions of temporal risk in adequacy studies is to use a 2D heat map with the same axes, but where each hour is associated with a non-overlapping rectangle and the color of the rectangle indicates shortfall risk magnitude (e.g. ranging from white, no risk, to dark red, high risk).

Since we aim to capture the RA failure events over a 4 month period (June - September with 122 days) and the events we observe are restricted to a small fraction of the hours in this period, we find that the contour representation gives a smeared out picture of the RA failure occurrence. Figure 7 below shows the comparison of the two representations, and we think the star plots illustrate the RA failure events better. As such, while we appreciate the reviewer’s suggestion, we have stuck with the star plots.

Figure 7: Comparison of star plot vs contour plot to visualize RA profile.

Reviewer #2 (Remarks to the Author):

This paper addresses an important topic area on exploring the meteorological drivers for power system resource adequacy (RA) performance. The authors map the system RA performance with different meteorological patterns for the western United States, and analyze the linkage of the two under alternative variable renewable penetration levels. Overall, the paper provides novel insights into an important research question. The manuscript is well-written and the modeling methods are clear.

Thank you very much for your detailed comments and positive statements regarding our paper. We addressed each of your comments below.

Some suggested changes and concerns are summarized below:

2.1. Definition and the use of “RA”: the term “RA” in this paper is vaguely defined and can result in confusions. While the definition is cited from a NERC report, “RA” is generally used to indicate long-term planning-scale system reliability (as opposed to short-term operational reliability). Even though RA is linked more closely with operational aspects recently, the authors should clarify the definition and the use of RA they are discussing in this paper.

Also, the authors should clarify what aspects of “RA” they are referring to when mentioning “RA”. For example, on page 3 line 118, the research question part: circulation patterns do not drive “RA”, they can impact RA performance. Also, on page 3 line 134, “RA profile” is not a standard term, please consider adding further explanations to it or replace the term. Please check throughout the paper to make all the uses of “RA” terms more explicit.

Thank you for bringing this opportunity for clarification to our attention. We have added several points of clarification in our manuscript regarding how we are considering and quantifying RA.

In our introduction, we have added the following text after defining our research question to concretely define RA:

“We define resource adequacy (RA) as the ability of a power system to continually balance electricity supply and demand, and quantify RA on a probabilistic, hour-to-hour operational basis.”

Given that we are focusing on failures of resource adequacy, we have also updated the first research question to read (bold text indicates addition): “What large-scale circulation patterns drive regional resource adequacy **failures**?”

In the last paragraph of introduction, we have also specified the definition of an RA profile:

“From the RAM, we obtain a timeseries of loss of load probabilities (LOLPs) by hour of the year, which we refer to as the RA profile. This RA profile is a function of short-term

operations from the RAM. Hours with LOLPs greater than zero indicate a risk of an RA failure; we refer to these hours as "RA risk hours" or "risk hours". "

Along with explicit definitions of RA, RA profile, RA failures, and RA risk hours, we have ensured intentional use of these terms in the manuscript and removed any ambiguity, thanks to the reviewer's suggestion.

2.2. Contribution and application of the work: this paper seems to analyze how meteorological drivers impact "RA performance" given fixed resource mix, and does not consider how these RA performance issues can reversely impact investments, i.e., this is a one-way impact analysis. The work highlights the needs to incorporate these meteorological and operational impacts in the RA assessments to incentivize an adequate system supply, because essentially these assessments and the suggested higher risks should all be linked back to adjusting system investment. It also suggests purely based a stylized planning reserve margin value in planning and investment decisions are not sufficient to guarantee operational reliability. The work can also provide insights to short-term operation by mapping certain weather patterns with reliability risks, and therefore suggesting red flags in system operation when certain patterns show up in forecasts.

Your description of our research as a one-way impact analysis is, we think, correct. Specifically, we do not consider feedback from our RA results back into investment decisions. We instead have a one-way flow from investments to the RAM to quantification of resource adequacy.

We also agree with your assessment of the value of our paper: by mapping weather patterns to risks of RA failures, we can improve long-term planning, mid-term scheduling, and short-term operations of power systems. We have better elucidated the values of our paper in these applications as follows:

"The spatial coverage of these large-scale atmospheric circulation patterns makes them valuable analogues for surface meteorology over large geographic regions. Using these synoptic drivers in planning and operations can benefit system operators when thinking about RA due to current and future systems' increasing dependence on generation over larger areas and interconnected balancing authorities."

2.3 As such, the authors should 1) when mention "RA", clarify that these are performance or levels or something else. For example, on page 2 line 85 (and other places), "RA failures" indicates "performance failure" rather than "investment failure" because the system already meets the identified planning reserve margin in the capacity expansion model; 2) modify the Introduction section to make the above point (one-way impact analysis) and the contribution more clear, and rephrase key sentences such as (among others) "this paper aims to better understand the meteorological drivers of RA and how investment decisions in renewable energy affect those drivers" (page 2, line 70) – meteorological driver of RA performance, and investment decisions should not be a static portfolio to affect drivers.

Thank you for these suggestions, all of which we agree with.

We have clarified RA failures refers to short-term performance failures, not investment failures, via the following text:

“RA failures, i.e., times where demand exceeds supply operationally at bulk power systems (BPS) level,, are often responsible for large-scale rolling outages”

We have also added a clarification regarding our one-way approach to the Introduction as follows:

“Using a one-way impact analysis which decides fleet investment, identifies resource adequacy failures, and finds meteorological drivers of these failures for increasing renewables penetrations, our research is the first to link weather patterns and power systems operations in the United States, and the first to characterize weather regimes driving RA failures.”

Finally, we have also flagged the value of future research in using our insights in a two-way analysis, particularly under future climates by finding periods of interest and then using them in more fine grained models. We have conveyed this with additions to our Discussion as follows:

“Our findings can improve this subsampling process by providing a physical basis for choosing the representative periods. Further, to make informed investment decisions and maintain system reliability in the future, system planning needs to use future meteorological data from climate projections and the physics based subsampling procedure can help here as well. Future climate projections from global climate models have lower spatial and temporal resolution than required by power system models. Incorporating this future climate data requires computationally costly downscaling. Our methods can reduce downscaling needs and associated costs by guiding selective downscaling of certain time periods of interest, e.g. time periods with high pressure anomalies in the Western US, to drive system planning and operation models.”

3. Better explanation of methods and assumptions justifications:

2.3.1 - Why choosing 0.005 as the threshold for “risk hours”?

In the original submission we had to use this threshold to drop some hours (noisy data) which appeared due to the sampling procedure and lack of storage. With the inclusion of storage and modified methods we do not require this criteria and use any hour with LOLP > 0 as a risk hour. We have clarified our methods as follows:

Section 3.3, line 771

“We refer to any hour with a LOLP > 0 to be a “risk hour””

2.3.2 - Examples of 12 weather regimes to illustrate what types of key weather phenomenon they represent (e.g., a major heat/cold wave event etc.), especially for WP 8, 9 and 10 ones due to their significant impacts.

In our results, we provide Figure 5 (copied below as Figure 8) and Figures SI 11,12,13, which provide composite surface-level meteorology corresponding to each weather regime for different weather years analyzed. We think this is the best way to indicate what types of weather phenomenon correspond to which regimes on average. To draw attention to this relationship earlier in our manuscript, we have added the following text:

“Each weather regime produces different surface weather patterns, e.g. high pressure anomalies in WPs 7 and 8 drive extreme heat events across the Western US, as later illustrated in our results.”

Figure 8: *Composites of surface temperature (A), surface solar radiation (B), and 100 m wind speeds (C) anomalies. The composites are constructed based on the hours from the 2016 extended summer belonging to each weather regime.*

2.3.4 - Relationships between Z500 anomalies data, surface meteorology data and temperature data used for RAM modeling: for example, are the temperature data for surface meteorology the same as the temperature used in RAM for temperature-dependent outage calculation?

Yes, the meteorological data that drives the CEM and RAM are the same data as used in the meteorology anomaly analyses. We state this in the paper as follows:

Section 3.5.2, line 810:

“The weather data used for surface meteorological anomalies and weather pattern identification for each weather year coincides with the weather data used to drive the power system models for the corresponding weather year.”

2.4. Page 13, line 565-574, the authors discussed the generator fleet adjustments to match the LOLH=2.4 requirement, such that the system has equivalent RA performance with the capacity expansion results. Why is this necessary? The planning reserve margin of 15% in the CEM is an approximation itself, and it is not necessarily equivalent to a LOLH=2.4 system. The LOLH value of the system mix from the CEM should be what comes from the RAM simulations. Manually adjusting the generators can create discrepancies between the two systems.

Thank you for this question. We agree that the PRM we enforce is an approximation of the reliability target, and resulting systems will not necessarily be equivalent to a LOLH=2.4 system. If we were only evaluating the resource adequacy of 1 system, we would not manipulate it. Here, however, we want to understand drivers of resource adequacy failures across renewable penetrations and weather years. This has two consequences for our analysis:

1. We want to have systems that have some probability of failure.
2. We want to be able to compare outcomes between systems.

To meet these conditions, we modify our systems output by the CEM so that all systems achieve an LOLH of roughly 2.4. That allows us to explore weather drivers of potential resource adequacy failures in each system, and also facilitates comparison between systems by controlling for the overall reliability of each system.

We have added text better explaining why we take this approach, reflecting the above explanation, to our manuscript:

“Unlike our RAM, our CEM does not account for stochastic outages. Instead, the CEM aims to produce a resource adequate system by enforcing a planning reserve margin. To facilitate resource adequacy comparisons across future systems output by our CEM, our RAM adjusts the generation fleets in CAMX for each case we model so that each fleet’s annual resource adequacy achieves a target value. Specifically, the RAM iteratively adds or removes NGCC capacity in CAMX then calculates annual resource adequacy until the annual loss of load hours (LOLH = $\sum(\text{LOLP})$) is 2.4 in each case. This target value reflects the real-world 1-in-10 reliability standard widely adopted by utilities. Due to high computational time taken to obtain the RA profiles and a priori unknown number of addition/removal trials of NGCC capacity, the iterative procedure is performed with 50 Monte

Carlo samples at each stage. This means that the final fleets all do not have an exact LOLH = 2.4, but vary between LOLH = 2 to LOLH = 2.6.”

2.5. The RAM analysis is an 8760 hourly simple Monte Carlo simulation, whereas in the real world the system operation can have multi-day-ahead to day-ahead or intra-day forecasts and operational adjustments if the operator foresees the weather events. Please add discussion on how ignoring these impacts may impact your results.

We appreciate the reviewer raising this very important aspect. While the operator can make operational adjustments, the RAM works with the maximum available capacity, so an outage signal with high probability indicates higher likelihood that operational adjustments on the generation side may not help with this idealized fleet. But, with increasing flexible load capabilities in many systems, there is potential for operational adjustments on the demand side. To that end, we have added the following text to our manuscript:

“We do not consider the availability of flexible loads in our models, which can be an avenue for operational adjustments by the system operator to prevent RA failures. Incorporating these demand side changes could reduce the risk in hours with high failure susceptibility.”

Having said that, we have now included discussion to indicate how our findings itself can be used to augment the existing short-term forecasting usage in power system operation:

“At the operational level, system operators, utilities, power producers, and communities can use the short term forecasts at the days to weeks timescale and long-range probabilistic forecasting at season-to-season time scale to avoid scheduling maintenance and other related down times when these patterns are expected to occur.”

2.6. Other minor questions/comments:

2.6.1 - Page 2, line 80, this is the first time the term “GCM” occurs. Please add it, otherwise there is no reference of “GCM” term for following usage

This has been rectified with “general circulation models (GCMs)” added the first time it occurs.

2.6.2 - Figure 1 and 2: the figures are too small

We have updated the font size and resolution to make figure 1 more legible, but unable to split that figure as it illustrates the whole pipeline. We have split figure 2 into separate figures.

Reviewers' Comments:

Reviewer #1:

Remarks to the Author:

I greatly appreciate the authors' updates to the manuscript, and feel my previous concerns have either been addressed or convincingly rebutted. I particularly appreciate the addition of transmission constraints to the CEM and RAM, as well as the expanded discussion of how insights on key WP trends can be used to selectively target periods with specific climatic conditions for computationally-expensive GCM downscaling.

I only have one remaining comment, and one minor correction:

- It remains unclear to me (based on both the main text and the SI) how storage is dispatched in the RAM? Section 3.3 explains that residual demand is calculated before considering contributions from hydropower and storage, and then goes on to describe how hydropower dispatch is determined. But no explanation is given for if or how storage is deployed if shortfalls persists after adding the hydropower generation. The response letter mentions that "storage assets are independently dispatched for each Monte Carlo iteration", which I interpret as some form of chronological dispatch calculated for each of the 250 replications? If this is the case I might explicitly mention this in the introduction as the RA sampling is described as "non-sequential Monte Carlo", and chronological modeling and sequential Monte Carlo sampling are commonly assumed to go hand-in-hand. This may cause readers to assume that chronological storage dispatch was not performed.

- on line 459, EUE is mistakenly referred to as "effective unserved energy", rather than "expected unserved energy" ("expected unserved energy" is correctly used elsewhere in the manuscript)

Reviewer #2:

Remarks to the Author:

The authors have properly addressed my previous review comments and added additional modeling results by considering transmission in the modeling region. Two minor suggestions based on the responses that authors provided:

The response to comment #2.3 around the contribution of the research: the added explanation "Using a one-way impact analysis which decides fleet investment, identifies resource adequacy failures..." still contains some ambiguity. Please make it more explicit that the process to determine investments/future power systems only rely on a simplified IRM representation.

The response to comment #2.4: please clarify explicitly that the resulting future systems from CEMs are meant to provide a reference/approximation of the systems you'll compare in RAMs and won't be used directly.

Dear Reviewers,

Thank you for the opportunity to again revise our manuscript, and for the feedback we received. In response to reviewer comments, we have made changes to our manuscript. We think these changes are fully responsive to the reviewer's concerns, and have improved our paper's quality.

Reviewer #1 (Remarks to the Author):

I greatly appreciate the authors' updates to the manuscript, and feel my previous concerns have either been addressed or convincingly rebutted. I particularly appreciate the addition of transmission constraints to the CEM and RAM, as well as the expanded discussion of how insights on key WP trends can be used to selectively target periods with specific climatic conditions for computationally-expensive GCM downscaling.

I only have one remaining comment, and one minor correction:

1.1 It remains unclear to me (based on both the main text and the SI) how storage is dispatched in the RAM? Section 3.3 explains that residual demand is calculated before considering contributions from hydropower and storage, and then goes on to describe how hydropower dispatch is determined. But no explanation is given for if or how storage is deployed if shortfalls persists after adding the hydropower generation. The response letter mentions that "storage assets are independently dispatched for each Monte Carlo iteration", which I interpret as some form of chronological dispatch calculated for each of the 250 replications? If this is the case I might explicitly mention this in the introduction as the RA sampling is described as "non-sequential Monte Carlo", and chronological modeling and sequential Monte Carlo sampling are commonly assumed to go hand-in-hand. This may cause readers to assume that chronological storage dispatch was not performed.

Thank you for pointing out this oversight. Your interpretation is correct - storage dispatch is performed chronologically. We have modified the paragraph describing the RAM in our introduction by making the following additions (in bold):

"We use empirically-derived temperature-dependent forced outage rates for NGCC and hydropower facilities, constant outage rates for other generators, **and do not account for outages in storage units. Storage assets are dispatched on a chronological hourly basis within the RA model within each Monte Carlo iteration after dispatching all the other generators using a greedy dispatch policy.**"

1.2 On line 459, EUE is mistakenly referred to as "effective unserved energy", rather than "expected unserved energy" ("expected unserved energy" is correctly used elsewhere in the manuscript)

Thank you for pointing this out, we have now corrected it.

Reviewer #2 (Remarks to the Author):

The authors have properly addressed my previous review comments and added additional modeling results by considering transmission in the modeling region.

Two minor suggestions based on the responses that authors provided:

2.1 The response to comment #2.3 around the contribution of the research: the added explanation “Using a one-way impact analysis which decides fleet investment, identifies resource adequacy failures...” still contains some ambiguity. Please make it more explicit that the process to determine investments/future power systems only rely on a simplified IRM representation.

Thank you for this comment. Our capacity expansion model relies on the planning reserve margin and supply and demand balance constraints as resource adequacy representations, which you are correct in stating are simplified representations of a full stochastic resource adequacy representation as employed in our resource adequacy model. For that reason, we use the RAM to modify fleet thermal generation capacity so that each system meets a LOLH=2.4 target. Hence, we supplement our CEM with our RAM to create future power systems for analysis. We have modified the sentence highlighted here to read (bold indicates the change made in this revision):

“Using a one-way impact analysis that decides fleet investment **to meet the standard resource adequacy target (1 day in 10 years)**, identifies resource adequacy failures, and finds meteorological drivers of these failures for increasing renewables penetrations, our research is the first to link weather patterns and power systems operations in the United States, and the first to characterize weather regimes driving RA failures.”

2.2 The response to comment #2.4: please clarify explicitly that the resulting future systems from CEMs are meant to provide a reference/approximation of the systems you’ll compare in RAMs and won’t be used directly.

Thank you for this comment. To clarify that the fleets from the CEM are not the final analysis fleet, we have added the following text at the end of the first paragraph in section 3.2:

“Thus, the fleets generated from the CEM form a basis for creating the final fleets used in our analysis. These final fleets are obtained after the RAM adds or removes thermal generators.”